# COSA: Concatenated Sample Pretrained Vision-Language Foundation Model

**Sihan Chen**[1,2,*]  **Xingjian He**[2]  **Handong Li**[1,2]  **Xiaojie Jin**[3,✉]  **Jiashi Feng**[3]  **Jing Liu**[1,2,✉]

[1]School of Artificial Intelligence, University of Chinese Academy of Sciences
[2] Institute of Automation, Chinese Academy of Science [3]Bytedance Inc.
`{sihan.chen, xingjian.he, jliu}@nlpr.ia.ac.cn`
`lihandong2023@ia.ac.cn,{jinxiaojie,jshfeng}@bytedance.com`

## Abstract

Due to the limited scale and quality of video-text training corpus, most vision-language foundation models employ image-text datasets for pretraining and primarily focus on modeling visually semantic representations while disregarding temporal semantic representations and correlations. To address this issue, we propose COSA, a **CO**ncatenated **SA**mple pretrained vision-language foundation model. COSA can jointly model visual contents and event-level temporal cues using only image-text corpora. We achieve this by sequentially concatenating multiple image-text pairs as inputs for pretraining. This transformation effectively converts existing image-text corpora into a pseudo video-paragraph corpus, enabling richer scene transformations and explicit event-description correspondence. Extensive experiments demonstrate that COSA consistently improves performance across a broad range of semantic vision-language downstream tasks, including paragraph-to-video retrieval, text-to-video/image retrieval, video/image captioning and video QA. Notably, COSA achieves state-of-the-art results on various competitive benchmarks. Code and model are released at `https://github.com/TXH-mercury/COSA`.

## 1 Introduction

Image-text and video-text pretraining models have garnered increasing attention in recent years due to their strong capabilities in bridging the domains of vision and language. These models contribute significantly to various vision-language tasks such as cross-modality retrieval, vision captioning, and visual question answering (VQA). In the early stages, these models were independently trained using distinct cross-modality corpora, network architectures, and training objectives. However, considering that a video can be perceived as a sequence of multiple images (Lei et al., 2021), and an image can be seen as a frozen video (Bain et al., 2021), most researchers lean towards training unified foundation models using corpora from both domains, employing sparsely sampled frames as the video representation.

Nevertheless, the majority of large-scale vision-language foundation models are usually trained solely on image-text corpora, disregarding the joint modeling of images and videos. This limitation stems from the restricted scale of video-text corpora, which is associated with the costly processes of video uploading, storage, and downloading: currently, open-sourced web-crawled video-text corpora, such as WebVid10M (Bain et al., 2021), are still two orders of magnitude smaller than its image-text counterpart (LAION-2B (Schuhmann et al., 2021)), which implies the dominance of image-text data in joint modeling. On the other hand, image-text pretraining can also bring benefits to video-related tasks. In fact, some large-scale image-text foundation models (Radford et al., 2021; Wang et al., 2022b) have achieved competitive or even superior performance on video tasks compared to dedicated video-language pretraining models.

---

* Sihan Chen did this work when interning at Bytedance Inc.
✉ Corresponding authors

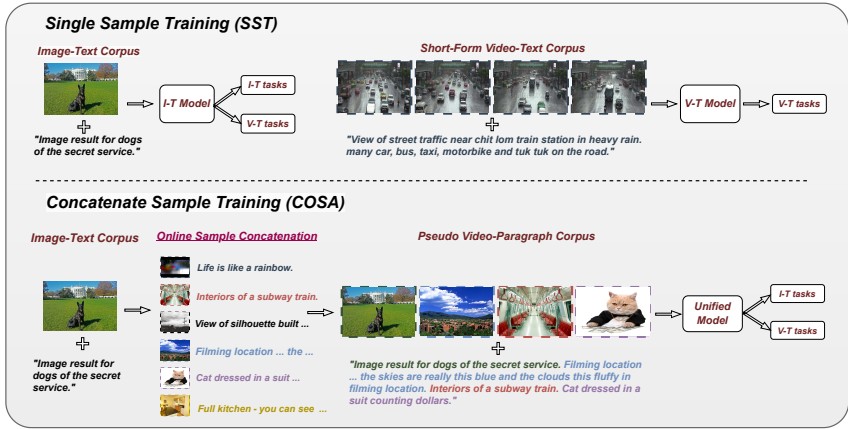

Figure 1: Visualizations of the traditional image-text and video-text model pretraining pipeline and proposed unified foundation model COSA, which transforms the image-text corpus into a synthetic video-text corpus through online sample concatenation.

Despite the fact that image-text foundation models have acquired a broad range of semantic correlations between visual concepts and language, they totally overlook temporal contexts within videos, which are crucial for video-language reasoning. This raises the question that *whether we can utilize large-scale image-text corpora to train a unified foundation model capable of capturing both static and temporal information during the pretraining phase.*

Towards this goal, we propose a simple yet effective approach: training a foundation model with concatenated samples. As illustrated in Figure 1, instead of modeling independently modeling cross-modality samples via traditional single sample training (SST), COSA models the semantic and event-level temporal information among multiple samples by conducting online sample concatenation to transform image-text corpora into video-paragraph corpora, and thus further enhance the learning of both image-language and video-language representations. Specifically, at each training step, for every image-text training sample, we randomly select some other samples from current batch and concatenate them (both images and texts) together. Ablation studies have demonstrated that model trained with COSA can evidently outperform model trained with SST on 12 benchmarks including both video-text and image-text tasks. We attribute it to two main reasons: 1) Concatenated samples build connections between visual concepts and words within a sample group, which is more difficult than build them within single sample, due to that both negative unmatched visual concepts and textual words become more, and thus is helpful for learn more discriminative representations. 2) In comparison to current video-text corpora (Bain et al., 2021), the on-the-fly concatenated corpus offers richer scene transformations, reduced vision redundancy due to sampling randomness, and longer temporally fine-grained captions that describe every frame in sequential order, which encourages models to learn better event-level temporal understanding capabilities and thus further improve performance on video-text tasks.

Enhanced by COSA, we deliver a unified vision-language foundation model. It possesses a simple architecture, takes concatenated samples as input during pretraining with training objectives modifid to build connections among multiple cross-modality samples. Extensive experiments have shown that COSA model is capable of handling both discriminative and generative tasks such as cross-modality retrieval, captioning, question answering, and achieves new state-of-the-art results across a wide range of benchmarks under different model scale settings.

## 2 RELATED WORK

**Temporal Learning in Video-Language Pretraining (VidLP).** Temporal information is the essence that distinguishes videos from images. Early methods aimed to enhance temporal learning in VidLP by either employing models pretrained on action classification tasks as vision backbones (Bain et al., 2021; Li et al., 2022a; Fu et al., 2021; 2022) or designing specific training objectives such as frame order modeling (FOM) (Li et al., 2020; Zellers et al., 2021) or masked video modeling (MVM) (Fu et al., 2021; 2022). We divide temporal learning into two aspects, namely **event-level** and **action-level** temporal modeling, in which the former emphasize on the differences and relations across multiple events/scenes, while the latter concentrates the temporal relations of consecutive frames in single event/scene.

Figure 2: Model architecture and training objectives of COSA. Circles and squares in the figure represent global and patch features, respectively.

However, Singularity (Lei et al., 2022) reveals that temporal modeling in pretraining may not be necessary, as sampling a single frame can yield comparable performance to utilizing multiple frames. They attribute this to the static appearance bias in video-language downstream benchmarks. We hypothesize that another contributing factor lies in the limitation of current video-text corpus (WebVid-2.5M (Bain et al., 2021)) for both action-level and event-level temporal learning. In terms of action-level temporal learning, the corpus contains fewer annotated actions compared to recognition datasets like Kinetics-400 (Kay et al., 2017), and there is significant caption redundancy as actions are often associated with several predicates rather than the entire sentence. For event-level temporal learning, the corpus is typically constrained to a single scene and description, whereas rich scene transformations and event-sentence correspondence are crucial. To address these limitations, OmniVL (Wang et al., 2022d) extends the idea of UniCL (Lu et al., 2022) to the video domain and incorporates Kinetics-400 into the training corpus. HiTeA (Ye et al., 2022) tackles caption redundancy through specific fine-grained alignment designs. Compared to above two methods which target at enhancing action-level temporal learning, LF_VILA (Sun et al., 2022) constructs a long-form dataset from the large-scale HD_VILA_100M (Xue et al., 2022a) dataset to enhance event-level temporal learning. In contrast to the aforementioned methods, this work proposes concatenated sample learning to dynamically compose pseudo video-paragraph corpora from existing image-text or video-text corpus, and simultaneously enhance image-text and **event-level** temporal video-text learning.

**Unified Vision-Language Foundation Model.** The Frozen model (Bain et al., 2021) was the first to leverage a mixture of image-text (CC3M (Sharma et al., 2018)) and video-text corpus (WebVid2.5M (Bain et al., 2021)) for training VidLP models by sparsely sampling the video corpus and treating the image corpus as frozen videos. Subsequent VidLP methods have followed this route to support both image-text and video-text tasks, with advanced vision backbones (Fu et al., 2021; 2022; Ye et al., 2022) or specially designed training objectives (Li et al., 2022a; Ye et al., 2022; Ge et al., 2022b).

Regarding large-scale unified vision-language foundation models, they can be classified into two types: co-training methods and two-stage training methods. Co-training methods train on both image and video corpora simultaneously and a typical representative method is Flamingo (Alayrac et al., 2022). Two-stage training methods first train models on image-text corpora and then adapt them for video-text tasks. Image-text models can be trained via contrastive learning (Radford et al., 2021; Jia et al., 2021), generative learning (Wang et al., 2022b; Chen et al., 2022; Wang et al., 2021), or a combination of both them(Yu et al., 2022). Regarding adapting image-text models to video-text ones, CLIP-based models (Luo et al., 2022; Xue et al., 2022b) employ temporal fusion transformers and frame-word fine-grained alignment learning to enable text-to-video retrieval. GIT (Wang et al., 2022b) directly finetune models by concatenating multiple frame features as input to the text decoder and achieves state-of-the-art performance on most video captioning benchmarks. VideoCoCa (Yan et al., 2022) adapts the CoCa model (Yu et al., 2022) through continuous learning on video-language corpora. MaMMUT (Kuo et al., 2023) applies a similar temporal perceiving capability enhancement during finetuning as TubeViT (Piergiovanni et al., 2022). In comparison to the above methods, COSA simultaneously model image-text and video-text learning during the pretraining stage, with image-text corpora utilized only.

## 3 METHOD

### 3.1 SIMPLE MODEL ARCHITECTURE

COSA adopts a simple architecture, consisting of a vision encoder (ViT (Dosovitskiy et al., 2020)) and a text encoder (BERT (Devlin et al., 2018)), which facilitates both single-modality encoding and cross-modality interaction. Specifically, we introduce additional cross-attention layers between the self-attention and feed-forward layers of BERT. These cross-attention layers are activated during the forward pass for cross-modality understanding or generation, while deactivated during the forward

Table 1: Model configurations of COSA. CC14M is a combination of CC3M (Sharma et al., 2018), CC12M (Changpinyo et al., 2021), COCO (Lin et al., 2014), SBU (Ordonez et al., 2011), and VG (Krishna et al., 2017b) datasets. The largest 1.2B COSA model is trained with a data bootstrapping strategy inspired by BLIP (Li et al., 2022b), and captions in training corpora (marked with '*') are generated by us through an additionally trained captioner, more details can be found in Appendix. The 400M image-text pairs used in CLIP and LAION-400M (Schuhmann et al., 2021) used in EVAClip are also included in the training examples statistics. LAION-102M is a randomly sampled subset from LAION-400M.

| Model | Vision Encoder | Param | Training corpora | Example | Step |
|---|---|---|---|---|---|
| COSA-B | Swin-B(Liu et al., 2021) | 251M | CC3M+WebVid2.5M | 5M | 100K |
| COSA-B | Swin-B(Liu et al., 2021) | 251M | CC14M+WebVid2.5M | 17M | 160K |
| COSA-L | CLIP/ViT-L/14(Radford et al., 2021) | 468M | CC14M+WebVid2.5M | 417M | 160K |
| COSA | EVAClip/VIT-G/14(Sun et al., 2023) | 1.2B | CC14M*+LAION-102M* | 415M | 150K |

pass for single-modality encoding. The text encoder can serve as a single-modality encoder, cross-modality encoder and decoder.

## 3.2 CONCATENATED SAMPLE TRAINING

**Online Sample Concatenation.** As illustrated in Figure 2, unlike traditional foundation models that take a single image-text pair or frames-text pair as input (SST, single sample training), COSA employs an online transformation to convert them into a pseudo video-paragraph corpus. To illustrate this process, let's consider the image-text corpus. Given a batch containing $n_b$ image-text samples, for the $i$-th sample $S_i$ with its image and text denoted as $I_i$ and $T_i$ respectively, we randomly select $n_c$ ($n_c = 3$ by default) other samples from the current batch and group them together. Consequently, the single sample $(I_i, T_i)$ is transformed into a grouped sample $(I_{group} = [I_i, I_{j1}, ..., I_{jn_c}], T_{group} = [T_i, T_{j1}, ..., T_{jn_c}])$. Subsequently, we concatenate both the images and texts in the same order to establish their event-level temporal correspondence. Specifically, for the grouped images, we individually feed them into the vision encoder, incorporate learnable temporal embeddings into the output features, and concatenate them along the sequential dimension. Regarding the grouped texts, we directly concatenate them into a paragraph where the temporal relations can be encoded by BERT's vanilla position embedding layer. Through the process of in-order sample concatenation and the use of position embeddings, the model can learn explicit scene-level temporal alignment. This concatenation procedure is applied to all samples in the batch. COSA can also be applied on video-text corpus and in this scenario, we sample a single frame from each video and perform the same concatenation process.

The concatenated images can be viewed as a series of snapshots from multiple clips within a pseudo video, where each clip captures a different scene accompanied by its corresponding descriptions. It is noted that these pseudo videos exhibit rapid scene changes with reduced visual redundancy but little semantic coherence due to the random sampling process. However, our experiments demonstrate that the cost of sacrificing clip coherence is negligible, and random sampling yields superior results compared to other settings where clip relevance is intentionally enforced.

**Training Objectives.** We employ four training objectives, namely Image-Text Contrastive (ITC), Image-Text Matching (ITM), Masked Language Modeling (MLM), and Generative Modeling (GM), to enhance the model's cross-modality alignment, understanding, and generation capabilities. These objectives are modified to accommodate the use of concatenated samples as input, as opposed to single samples.

*Concatenated Image-Text Contrastive (CITC).* The pseudo video frames are individually passed through the vision encoder, and the average pooling of the output [CLS] token features serves as the global representation of the pseudo videos. The [CLS] token feature from BERT is utilized as the paragraph representation. A bi-directional contrastive loss is employed to bring paired concatenated samples closer together while pushing away unpaired ones.

*Concatenated Image-Text Matching (CITM).* This task requires the model to determine whether a pseudo video and a paragraph correspond to each other. Hard negative mining is used following the approach in (Li et al., 2021a). The [CLS] token feature from the multimodal encoder (text encoder) is used to compute a binary classification loss through a two-layer MLP.

Table 2: Performance comparison on text-to-video retrieval benchmarks. For fair comparison, evaluation results before employing post-processing methods such as dual softmax (Cheng et al., 2021) are compared. Recall@1,5,10 are used as evaluation metrics.

| Method | Example | MSRVTT | | | DiDeMo | | | LSMDC | | | ActivityNet | | |
|---|---|---|---|---|---|---|---|---|---|---|---|---|---|
| | | R@1 | R@5 | R@10 | R@1 | R@5 | R@10 | R@1 | R@5 | R@10 | R@1 | R@5 | R@10 |
| ClipBert(Lei et al., 2021) | 5.4M | 22.0 | 46.8 | 59.9 | 20.4 | 48.0 | 60.8 | - | - | - | 21.3 | 49.0 | 63.5 |
| Frozen(Bain et al., 2021) | 5M | 31.0 | 59.5 | 70.5 | 34.6 | 65.0 | 74.7 | 15.0 | 30.8 | 39.8 | - | - | - |
| BridgeFormer(Ge et al., 2022a) | 5M | 37.6 | 64.8 | 75.1 | 37.0 | 62.2 | 73.9 | 17.9 | 35.4 | 44.5 | - | - | - |
| MILES(Ge et al., 2022b) | 5M | 37.7 | 63.6 | 73.8 | 36.6 | 63.9 | 74.0 | 17.8 | 35.6 | 44.1 | - | - | - |
| OA-Trans(Wang et al., 2022c) | 5M | 35.8 | 63.4 | 76.5 | 34.8 | 64.4 | 75.1 | 18.2 | 34.3 | 43.7 | - | - | - |
| Clover(Huang et al., 2022) | 5M | 38.6 | 67.4 | 76.4 | 45.1 | 74.3 | 82.2 | 22.7 | 42.0 | 52.6 | - | - | - |
| VIOLETv2(Fu et al., 2022) | 5M | 37.2 | 64.8 | 75.8 | 47.9 | 76.5 | 84.1 | 24.0 | 43.5 | 54.1 | - | - | - |
| LF-VILA(Sun et al., 2022) | 8.5M | - | - | - | 35.0 | 64.5 | 75.8 | - | - | - | 35.3 | 65.4 | - |
| **COSA-B(Ours)** | **5M** | **42.2** | **69.0** | **79.0** | **57.8** | **80.6** | **87.9** | **27.3** | **46.7** | **55.2** | **55.6** | **80.7** | **89.1** |
| OmniVL(Wang et al., 2022d) | 17M | 47.8 | 74.2 | 83.8 | 52.4 | 79.5 | 85.4 | - | - | - | - | - | - |
| HiTeA(Ye et al., 2022) | 17M | 46.8 | 71.2 | 81.9 | 56.5 | 81.7 | 89.7 | 28.7 | 50.3 | 59.0 | 49.7 | 77.1 | 86.7 |
| SINGULARITY(Lei et al., 2022) | 17M | 41.5 | 68.7 | 77.0 | 53.9 | 79.4 | 86.9 | - | - | - | 47.1 | 75.5 | 85.5 |
| VINDLU-B(Cheng et al., 2022) | 17M | 45.3 | 69.9 | 79.6 | 59.2 | 84.1 | 89.5 | - | - | - | 54.4 | 80.7 | 89.0 |
| LAVENDER(Li et al., 2022c) | 30M | 40.7 | 66.9 | 77.6 | 53.4 | 78.6 | 85.3 | 26.1 | 46.4 | 57.3 | - | - | - |
| **COSA-B(Ours)** | **17M** | 46.9 | 72.1 | 81.3 | **64.1** | **86.1** | **90.6** | **31.2** | **50.9** | 57.8 | **59.3** | **83.8** | **90.9** |
| All-in-one(Wang et al., 2022a) | 138M | 37.9 | 68.1 | 77.1 | 32.7 | 61.4 | 73.5 | 22.4 | 53.7 | 67.7 | - | - | - |
| CLIP4Clip(Luo et al., 2022) | 400M | 44.5 | 71.4 | 81.6 | 43.4 | 70.2 | 80.6 | 22.6 | 41.0 | 49.1 | 40.5 | 72.4 | - |
| X-CLIP(Ma et al., 2022) | 400M | 49.3 | 75.8 | 84.8 | 47.8 | 79.3 | - | 26.1 | 48.4 | 46.7 | 46.2 | 75.5 | - |
| mPLUG-2(Xu et al., 2023) | 417M | 53.1 | 77.6 | 84.7 | 56.4 | 79.1 | 85.2 | 34.4 | 55.2 | 65.1 | - | - | - |
| UMT-L(Li et al., 2023b) | 425M | **58.8** | **81.0** | **87.1** | 70.4 | **90.1** | **93.5** | **43.0** | 65.5 | **73.0** | 66.8 | **89.1** | 94.9 |
| VALOR-L(Chen et al., 2023) | 433.5M | 54.4 | 79.8 | 87.6 | 57.6 | 83.3 | 88.8 | 31.8 | 52.8 | 62.4 | 63.4 | 87.8 | 94.1 |
| CLIP-VIP(Xue et al., 2022b) | 500M | 54.2 | 77.2 | 84.8 | 50.5 | 78.4 | 87.1 | 29.4 | 50.6 | 59.0 | 53.4 | 81.4 | 90.0 |
| **COSA-L(Ours)** | **417M** | 54.4 | 77.2 | 84.7 | 68.3 | 88.1 | 91.7 | 38.6 | 59.1 | 67.4 | 66.8 | 87.6 | 93.9 |
| **COSA(Ours)** | **415M** | 57.9 | 79.6 | 86.1 | **70.5** | 89.3 | 92.4 | 39.4 | 60.4 | 67.7 | **67.3** | 89.0 | **95.0** |

*Concatenated Masked Language Modeling (CMLM).* We randomly mask 15% of the tokens in the paragraphs and employ BERT's vanilla prediction layer to reconstruct the masked tokens given the context of the pseudo videos.

*Concatenated Generation Modeling (CGM).* We randomly mask 60% of the tokens in the paragraphs and utilize the same prediction layer as in CMLM to reconstruct the masked tokens in the context of the pseudo videos. CGM incorporates causal attention masks in the self-attention layers of BERT to prevent information leakage and enhance the model's capabilities for text generation.

In addition to the four training objectives that utilize concatenated samples as input, we also include two vanilla objectives (ITC and ITM) that take raw single samples as input to enhance the model's ability to process single samples.

## 4 EXPERIMENTS

### 4.1 IMPLEMENTATION DETAILS

We train COSA models using the PyTorch framework and 64 Tesla V100 cards. To ensure a fair comparison with state-of-the-art video-language pretraining models and vision-language foundation models, we train four model variants with different parameter and data sizes, as illustrated in Table 1. All models utilize BERT-Base as the text encoder. The initial learning rate is set to 1e-4, and a 10% warm-up strategy with a linear decay schedule is employed. The batch size is set to 2048. For ablation studies, we utilize the frozen CLIP/ViT-B/16 (Radford et al., 2021) as the vision encoder for efficiency purposes, and the models are trained on CC3M for 30K steps with a batch size of 1024, unless specified otherwise. Six training losses are equally weighted. For pretraining and video-text tasks finetuning, we use 224 image resolution, while for image-text tasks finetuning, we use higher resolution, which is presented together with Other details in Appendix.

### 4.2 COMPARISON TO STATE-OF-THE-ART

**Text-to-Video Retrieval.** We evaluate text-to-video retrieval on four benchmarks, including MSRVTT (Xu et al., 2016), DiDeMo (Anne Hendricks et al., 2017), LSMDC (Rohrbach et al., 2017), and ActivityNet (Krishna et al., 2017a). ITC and ITM are used as the training objectives. During testing, we rank all candidates using similarity scores computed by ITC and then rerank the top-50 candidates via ITM. As shown in Table 2, COSA-B (5M) outperforms previous models of similar scale by a significant margin on all four benchmarks, especially for the paragraph-to-video retrieval benchmarks (DiDeMo and ActivityNet). This demonstrates the effectiveness of concatenated sample training. It is noted that LF-VILA(Sun et al., 2022) is customized designed for paragraph-to-video retrieval with

Table 3: Performance comparison on open-ended video QA benchmarks. Accuracy is used as the evaluation metric.

| Method | Example | MSR | MSVD | TGIF | ANet |
|---|---|---|---|---|---|
| ClipBERT(Lei et al., 2021) | 5.4M | 37.4 | - | 60.3 | - |
| ALPRO(Li et al., 2022a) | 5M | 42.1 | 45.9 | | - |
| VIOLETv2(Fu et al., 2022) | 5M | 44.5 | **54.7** | 72.8 | - |
| Clover(Huang et al., 2022) | 5M | 43.9 | 51.9 | 71.4 | - |
| **COSA-B(Ours)** | **5M** | **46.2** | 54.3 | **73.4** | **46.5** |
| OmniVL(Wang et al., 2022d) | 17M | 44.1 | 51.0 | | |
| HiTeA(Ye et al., 2022) | 17M | 45.9 | 55.3 | 73.2 | 46.4 |
| SINGULARITY(Lei et al., 2022) | 17M | 43.5 | - | - | 43.1 |
| VINDLU-B(Cheng et al., 2022) | 17M | 43.8 | - | - | 44.6 |
| LAVENDER(Li et al., 2022c) | 30M | 45.0 | **56.6** | 73.5 | - |
| JustAsk(Yang et al., 2021) | 69M | 41.5 | 46.3 | - | 38.9 |
| **COSA-B(Ours)** | **17M** | **46.9** | 55.5 | **75.0** | **47.3** |
| MERLOT(Zellers et al., 2021) | 180M | 43.1 | | 69.5 | 41.4 |
| All-in-one(Wang et al., 2022a) | 228.5M | 46.8 | 48.3 | 66.3 | - |
| FrozenBiLM(Yang et al., 2022) | 410M | 47.0 | 54.8 | 68.6 | 43.2 |
| mPLUG-2(Xu et al., 2023) | 417M | 48.0 | 58.1 | 75.4 | - |
| UMT-L(Li et al., 2023b) | 425M | 47.1 | 55.2 | - | 47.9 |
| VALOR-L(Chen et al., 2023) | 433.5M | 49.2 | 60.0 | 78.7 | 48.6 |
| InternVideo(Wang et al., 2022g) | 646M | 47.1 | 55.5 | 72.2 | - |
| GIT(Wang et al., 2022b) | 1.7B | 43.2 | 56.8 | 72.8 | - |
| Flamingo(80B)(Alayrac et al., 2022) | 2.3B | 47.4 | - | - | - |
| VideoCoCa (2.1B)(Yan et al., 2022) | 4.8B | 46.0 | 56.9 | - | - |
| GIT2 (5.1B)(Wang et al., 2022b) | 12.9B | 45.6 | 58.2 | 74.9 | - |
| **COSA-L(Ours)** | **417M** | 48.8 | 58.6 | 77.6 | 49.2 |
| **COSA(Ours)(1.2B)** | **415M** | **49.2** | **60.0** | **79.5** | **49.9** |

Table 4: Performance comparison on video captioning benchmarks. BLEU@4 (Papineni et al., 2002) (B@4) and CIDEr (Vedantam et al., 2015) (C) are used as metrics. Results of models utilizing additional modalities such as audio (Chen et al., 2023) and subtitle (Tang et al., 2021; Li et al., 2021b) are denoted by dashed color. Following (Tang et al., 2021; Chen et al., 2023; Wang et al., 2022b), we employ SCST finetuning (Rennie et al., 2017) on the VATEX benchmark, and the corresponding results are marked with '*'.

| Method | Example | MSRVTT | | MSVD | | VATEX | | YouCook2 | | TVC | |
|---|---|---|---|---|---|---|---|---|---|---|---|
| | | B@4 | C | B@4 | C | B@4 | C | B@4 | C | B@4 | C |
| VIOLETv2(Fu et al., 2022) | 5M | - | 58.0 | - | 139.2 | - | - | - | - | - | - |
| HiTeA(Ye et al., 2022) | 5M | - | 62.5 | - | 145.1 | - | - | - | - | - | - |
| **COSA-B(Ours)** | **5M** | **48.1** | **64.3** | **66.9** | **146.0** | **40.5** | **72.6** | **7.8** | **100.9** | **15.6** | **59.2** |
| LAVENDER(Li et al., 2022c) | 30M | - | 60.1 | - | 150.7 | - | - | | - | - | - |
| **COSA-B(Ours)** | **17M** | **48.5** | **64.7** | **68.7** | **150.9** | **41.4** | **74.4** | **8.6** | **106.2** | **16.2** | **61.0** |
| VALUE(Li et al., 2021b) | 136M | - | - | - | - | - | 58.1 | 12.4 | 130.3 | 11.6 | 50.5 |
| CLIP4Caption++(Tang et al., 2021) | 400M | - | - | - | - | 40.6* | 85.7* | - | - | 15.0 | 66.0 |
| VALOR-L(Chen et al., 2023) | 433.5M | 54.4 | 74.0 | 80.7 | 178.5 | 45.6* | 95.8* | - | - | - | - |
| GIT(Wang et al., 2022b) | 1.7B | 53.8 | 73.9 | 79.5 | 180.2 | 41.6* | 91.5* | **10.3** | 129.8 | 16.2 | 63.0 |
| Flamingo(80B)(Alayrac et al., 2022) | 2.3B | - | - | - | - | - | - | - | 118.6 | - | - |
| GIT2 (5.1B)(Wang et al., 2022b) | 12.9B | **54.8** | **75.9** | **82.2** | 185.4 | 42.7* | 94.5* | 9.4 | 131.2 | 16.9 | 66.1 |
| **COSA-L(Ours)** | **417M** | 53.2 | 72.1 | 74.0 | 169.1 | **45.4** | 84.4 | 10.2 | 125.8 | 17.7 | 67.5 |
| **COSA(Ours)(1.2B)** | **415M** | 53.7 | 74.7 | 76.5 | 178.5 | 43.7* | **96.5*** | 10.1 | **131.3** | **18.8** | **70.7** |

specialized framework and corpus, but fall behinds COSA-B which takes a simple unified architecture as well as common corpora. In addition, COSA (17M) evidently surpasses VINDLU(Cheng et al., 2022) which have searched a best training recipe targeted at retrieval tasks. When compared to large-scale models, COSA achieves the best R@1 results on DiDeMo and ActivityNet datasets and comparable results on other benchmarks.

**Video QA.** We evaluate video QA on four benchmarks, including MSRVTT-QA (Xu et al., 2017), MSVD-QA (Xu et al., 2017), TGIF-QA (Li et al., 2016), and ActivityNet-QA (Yu et al., 2019), and formulate it as an open-ended generative task to predict answers with questions as prefixes. As shown in Table 3, COSA-B (5M) and COSA-B (17M) outperform all models of similar scale by a significant margin. Under the large-scale pretraining scenario, COSA achieves state-of-the-art accuracy on 4 benchmarks, using only 415M training data and 1.2B parameters.

**Video Captioning.** We evaluate video captioning on five benchmarks, including MSRVTT (Xu et al., 2016), MSVD (Chen & Dolan, 2011), VATEX (Wang et al., 2019), YouCook2 (Zhou et al., 2018), and TVC (Lei et al., 2020). The GM objective is used, and captions are autoregressively generated during inference. From the results in Table 4, we can see that COSA-B (5M) outperforms

Table 5: Performance comparison on text-to-image retrieval, image captioning, and image QA benchmarks. 'ZS' stands for zero-shot evaluation. CIDEr (C) and SPICE (S) are reported for captioning. SCST finetuning is employed on COCO caption, and corresponding results are marked with '*'.

| Method | Example | MSCOCO-Ret | | | Flickr30K-Ret (ZS) | | | MSCOCO-Cap | | VQAv2 | |
|---|---|---|---|---|---|---|---|---|---|---|---|
| | | R@1 | R@5 | R@10 | R@1 | R@5 | R@10 | C | S | dev | std |
| OFA(Wang et al., 2022e) | 18M | - | - | - | - | - | - | **154.9*** | 26.6* | 82.0 | 82.0 |
| BEiT-3(Wang et al., 2022f) | 21M | 67.2 | 87.7 | 92.8 | 81.5 | 95.6 | 97.8 | 147.6 | 25.4 | **84.19** | **84.03** |
| BLIP(Li et al., 2022b) | 129M | 65.1 | 86.3 | 91.8 | 86.7 | 97.3 | 98.7 | 136.7 | - | 78.25 | 78.32 |
| BLIP-2(Li et al., 2023a) | 129M | 68.3 | 87.7 | 92.6 | 89.7 | 98.1 | 98.9 | 145.8 | - | 82.19 | 82.30 |
| mPLUG-2(Xu et al., 2023) | 417M | 65.7 | 87.1 | 92.6 | - | - | - | 137.7 | 23.7 | 81.11 | 81.13 |
| VALOR(Chen et al., 2023) | 433.5M | 61.4 | 84.4 | 90.9 | - | - | - | 152.5* | 25.7* | 78.46 | 78.62 |
| Florence(Yuan et al., 2021) | 900M | 63.2 | 85.7 | - | 76.7 | 93.6 | - | - | - | 80.16 | 80.36 |
| GIT(Wang et al., 2022b) | 1.7B | - | - | - | - | - | - | 151.1* | 26.3* | 78.6 | 78.8 |
| SimVLM(Wang et al., 2021) | 1.8B | - | - | - | - | - | - | 143.3 | 25.4 | 80.03 | 80.34 |
| ALIGN(Jia et al., 2021) | 1.8B | 59.9 | 83.3 | 89.8 | 75.7 | 93.8 | 96.8 | - | - | - | - |
| Flamingo (80B)](Alayrac et al., 2022) | 2.3B | - | - | - | - | - | - | 138.1 | - | 82.0 | 82.1 |
| CoCa(2.1B)(Yu et al., 2022) | 4.8B | - | - | - | 80.4 | 95.7 | 97.7 | 143.6 | 24.7 | 82.3 | 82.3 |
| GIT2(5.1B)(Wang et al., 2022b) | 12.9B | - | - | - | - | - | - | 152.7* | 26.4* | 81.7 | 81.9 |
| **COSA(Ours)(1.2B)** | **415M** | **68.5** | **88.0** | **93.0** | **90.2** | **98.4** | **99.4** | 150.6* | **27.0*** | 80.46 | 80.54 |

Table 6: Experiment results of SST (baseline), COSA, and COSA variants on a broad range of video/image-language benchmarks. R@1, CIDEr, and Acc are reported for retrieval, captioning, and QA tasks, respectively.

| Exp | Video Ret | | | | Video Cap | | | Video QA | | Image Ret | Image Cap | |
|---|---|---|---|---|---|---|---|---|---|---|---|---|
| | DiDeMo | ActivityNet | MSRVTT | VATEX | YouCook | VATEX | TVC | MSRVTT | TGIF-QA | MSCOCO | MSCOCO | VIST |
| SST | 49.2 | 43.8 | 39.4 | 62.4 | 85.4 | 67.6 | 53.9 | 45.9 | 73.0 | 53.9 | 124.7 | 14.8 |
| COSA | **54.8** | **51.2** | 43.5 | 63.9 | **91.9** | **68.1** | **55.4** | 46.5 | 73.7 | **54.7** | **125.4** | **20.7** |
| COSA-copy | 43.5 | 39.3 | 42.3 | 60.8 | 84.6 | 65.8 | 52.0 | 45.4 | 72.2 | 51.6 | 122.7 | 11.9 |
| COSA-shuffle | 50.9 | 48.1 | **44.7** | **64.0** | 88.8 | 67.7 | 54.4 | **46.6** | **73.9** | 54.5 | 124.9 | 14.2 |

VIOLETv2, and HiTeA which utilizes a more advanced vision backbone (MViTv2 (Fan et al., 2021)). COSA-B (17M) surpasses LAVENDER, which is trained on 30M vision-text pairs. When compared to large-scale foundation models, COSA achieves comparable performances on the MSRVTT and MSVD datasets. Additionally, COSA outperforms GIT2, which specializes in captioning tasks, and achieves new state-of-the-art performances on both open-domain benchmarks (Wang et al., 2019) and domain benchmarks (Zhou et al., 2018; Lei et al., 2020) (cooking and TV), using only 24% of its parameters and 3.2% of the training examples. It is worth noting that both GIT2 and COSA do not use any video-text corpus in the pretraining stage, but COSA can generate online pseudo video-paragraph pairs. Furthermore, utilizing the vision modality alone, COSA achieves higher CIDEr scores than VALOR and CLIP4Caption++, which additionally incorporate audio and subtitle modalities.

**Image-Text Benchmarks.** We evaluate COSA on image-text benchmarks, including text-to-image retrieval on MSCOCO (Lin et al., 2014) and Flickr30K (Plummer et al., 2015) (zero-shot), captioning on MSCOCO, and image QA on VQAv2 (Goyal et al., 2017). As shown in Table 5, COSA surpasses BLIP-2 and BEiT-3, and achieves new state-of-the-art results on the MSCOCO and Flickr30K benchmarks. Additionally, COSA achieves the best SPICE score (27.0) on MSCOCO caption benchmark and comparable results on VQAv2 benchmark.

## 4.3 ABLATION STUDY

In this subsection, we conduct extensive ablation experiments to demonstrate the effectiveness of COSA when compared with SST and the method's robustness to different vision backbones and training corpora. We also explore the influence of training objectives, iterations, concatenation numbers, and sampling methods. Besides above mentioned benchmarks, we further evaluate the model on the Visual Storytelling Benchmark (VIST), which requires models to perceive the relationships among multiple images and describe them in sequential paragraphs, thus demanding strong narrative abilities.

**COSA vs SST.** As presented in Table 6, COSA consistently improves performance on 12 benchmarks of video-text as well as image-text tasks, with significant margins compared to the SST baseline.

Table 7: Experiment results of different training objectives for COSA. R, C, Q in the parentheses represent retrieval, captioning, and QA tasks, respectively.

| | ITC | ITM | MLM | GM | CITC | CITM | CMLM | CGM | DiD(R) | ANET(R) | MSR(R) | TVC(C) | TGIF(Q) |
|---|---|---|---|---|---|---|---|---|---|---|---|---|---|
| a | ✓ | ✓ | ✓ | ✓ | | | | | 49.2 | 43.8 | 39.4 | 53.9 | 73.0 |
| b | ✓ | ✓ | | | | | ✓ | ✓ | 53.0 | 50.6 | 42.8 | **55.7** | 73.7 |
| c | ✓ | ✓ | | | ✓ | ✓ | ✓ | ✓ | **54.8** | **51.2** | **43.5** | 55.4 | **73.7** |
| d | ✓ | ✓ | ✓ | ✓ | ✓ | ✓ | ✓ | ✓ | 52.3 | 49.9 | 42.3 | 55.5 | 73.6 |

Table 8: Experiment results of changing dataset and vision backbone (Enc).

| Exp | Enc | Data | #Frame | DiD(R) | ANET(R) | MSR(R) | YOU(C) | VIST(C) | TVC(C) | MSR(Q) |
|---|---|---|---|---|---|---|---|---|---|---|
| SST | Swin-B | CC3M | - | 35.3 | 30.4 | 34.7 | 91.2 | 14.3 | 51.6 | 44.6 |
| COSA | Swin-B | CC3M | - | **45.1** | **41.7** | **36.2** | **94.6** | **22.3** | **53.1** | **44.9** |
| SST | CLIP-B | WebVid2.5M | 1 | 43.6 | 40.2 | 40.7 | 89.3 | 12.2 | 53.5 | 46.0 |
| SST | CLIP-B | WebVid2.5M | 4 | 51.4 | 47.3 | 42.5 | 90.8 | 11.9 | 54.0 | 46.2 |
| COSA | CLIP-B | WebVid2.5M | 1 | **54.7** | **52.2** | **44.0** | **93.4** | 17.4 | **55.0** | **46.4** |

We further investigate two COSA variants named COSA-copy and COSA-shuffle. COSA-copy simply duplicates a single sample (image and text) multiple times and concatenates them, instead of performing cross-sample concatenation as in COSA. On the other hand, COSA-shuffle concatenates cross-samples from a batch, but with the captions randomly shuffled before concatenation to disrupt event-level temporal correspondence. From the results we can find that 1) Both COSA and COSA-shuffle can outperform SST with huge margins on all benchmarks, demonstrating the effectiveness of modeling vision-language connections within sample group instead of single sample. 2) Due to maintaining the same concatenation order for image and text, model can learn better event-level temporal correspondence between visual scenes and descriptions, and thus improve the performance on most benchmarks especially on paragraph-to-video retrieval benchmarks such as ActivityNet, whose video and caption are relatively longer and contain more complicated scenes. 3) COSA-copy achieves lower performance than SST on most benchmarks, demonstrating that the improvement of COSA over SST comes from multiple samples concatenation, instead of longer visual and text sequences.

**Training Objectives.** As depicted in Table 7, compared to the SST baseline, incorporating COSA while modifying the MLM and GM objectives to CMLM and CGM leads to evident improvements across all five benchmarks (b vs a). The default objectives (c) yield the best results for most benchmarks.

**Generality.** We further explore the generality of COSA with respect to different vision encoders and training corpora. The previous experiments employed CLIP-B as the vision backbone and were conducted on the CC3M dataset. In this experiment, we replace them with Swin Transformer(Liu et al., 2021) and WebVid2.5M, respectively. From the results in Table 8, it is evident that COSA consistently outperforms SST under different backbone and corpus settings. Additionally, we observe that sampling multiple frames for SSL can improve downstream performance (except for VIST), but is still much less effective when compared with proposed COSA.

**Equal samples seen.** In the previous experiments, models were trained for 30K steps for efficiency purposes. One might argues that the improvements of COSA could be attributed to faster convergence rather than cross-sample learning and introduced event-level temporal correspondence, as the model can see each sample multiple times per iteration. However, the results in Figure 4 demonstrate that as training iterations prolong, the performance gap between COSA and SST remains consistent, even when SST has already converged. Moreover, pretraining with COSA for 30K steps outperforms pretraining with SST for 120K steps on all benchmarks, which demonstrates COSA still outperforms SST with same samples seen and much less iterations.

**Concatenate Number.** Figure 3 shows the performance of models trained with different concatenate numbers ($n_c$) ranging from 0 (SST) to 4. Based on the results, we choose $n_c = 3$ as it consistently performs well across all benchmarks.

**Training Speed.** We have made statistics that transferring from SST to COSA averagely cost 0.9x more training time when four samples are concatenated. We think this is unavoidable cause the frames become more and sentences become longer, but affordable because we only need to pretrain once, and the finetuning, testing, inference speed of COSA and SST are totally the same.

Table 9: Experiment results of different sampling methods for online sample concatenation.

| Sampling | DiD(R) | ANet(R) | MSR(R) | VAT(R) | TVC(C) | VIST(C) | MSR(Q) |
|---|---|---|---|---|---|---|---|
| random | **54.8** | **51.2** | **43.5** | **63.9** | **55.4** | **20.7** | **46.5** |
| vision similarity | 53.7 | 50.9 | 43.0 | 63.6 | 55.1 | 20.4 | 46.4 |
| text similarity | 48.2 | 45.4 | 42.6 | 62.4 | 55.2 | 20.2 | 46.3 |

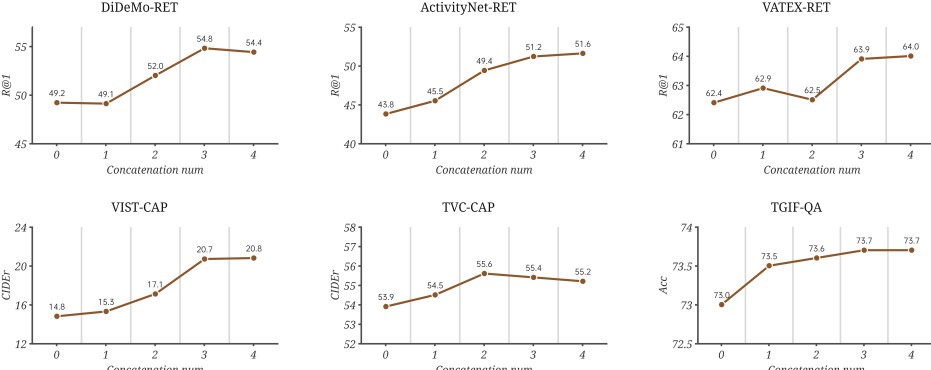

Figure 3: Experiment results of different concatenation numbers used in COSA.

**Sampling Choice.** By default, we randomly choose samples within a batch for grouping and concatenation. We also experimented with two alternative choices: selecting the top $n_c$ most similar samples based on the similarity matrix computed by either the image or text encoder output features. The results in Table 9 demonstrate the importance of reducing semantic relevance among grouped samples through random sampling. More explanations and visualizations could be found in Appendix.

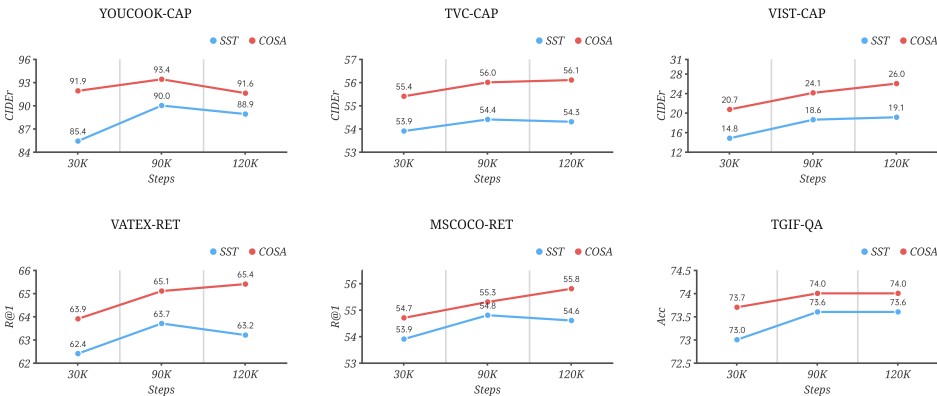

Figure 4: Experiment results of SST and COSA when training for more iterations.

## 5  CONCLUSION

This paper introduces COSA, a concatenated sample pretrained unified vision-language foundation model. COSA transforms image-text corpora into pseudo video-paragraph corpora, enabling explicit event-sentence correspondence and promoting event-level temporal learning in foundation models. Extensive ablation studies demonstrate the effectiveness of COSA in improving various downstream tasks, including video-language tasks and image-text tasks. COSA achieves state-of-the-art results on multiple benchmarks. However, it is important to note that the proposed COSA has only been proven effective in the vision-language pretraining field, and we hope that it can inspire further research on co-learning multiple samples in other AI communities.

ACKNOWLEDGMENTS

This work was supported by the National Science and Technology Major Project (No.2022ZD0118801), National Natural Science Foundation of China (U21B2043, 62206279).

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

# A   APPENDIX

## A.1   PRETRAINING SETTINGS

Table 10: Pretraining Settings for COSA models. '*' means that captions in those corpora has been replaced by descriptions generated by a separately trained captioner.

| Model | Vision Encoder | param | Sample | Training Corpus | Batch Size | Steps | Epoch |
|-------|---------------|-------|--------|-----------------|-----------|-------|-------|
| COSA-B | Swin-B | 251M | 5M | CC3M | 2048 | 50000 | 35 |
| | | | | WebVid2.5M | 2048 | 50000 | 41 |
| COSA-B | Swin-B | 251M | 17M | CC4M | 2048 | 60000 | 25 |
| | | | | CC12M | 2048 | 60000 | 12 |
| | | | | WebVid2.5M | 2048 | 40000 | 33 |
| COSA-L | CLIP-ViT-L | 468M | (17+400)M | CC4M | 2048 | 60000 | 25 |
| | | | | CC12M | 2048 | 60000 | 12 |
| | | | | WebVid2.5M | 2048 | 35000 | 29 |
| COSA | EVAClip-ViT-G | 1.2B | (15+400)M | CC4M | 2048 | 50000 | 20 |
| | | | | CC12M* | 2048 | 50000 | 10 |
| | | | | LAION-102M* | 2048 | 50000 | 1 |

The specific pretraining settings of all scales of COSA models are presented in Table 10. It is noted that the training corpus of the largest one (COSA) has been clean by a separately trained captioner. Specifically, the captioner takes the same model architecture of COSA and is pretrained with generative modeling loss (GM) only on the same corpus with original captions (CC4M, CC12M and LAION-102M) without concatenated samples applied. After pretraining, the captioner is finetuned with GM loss on a mixture of downstream of datasets including MSRVTT, VATEX, MSVD and MSCOCO. At last we use trained captioner to generate high-quality captions for CC12M and LAION-102M with top-10 sampling method.

## A.2   DOWNSTREAM DATASET DESCRIPTIONS

COSA is evaluated on a series of downstream datasets including MSRVTT, MSVD, LSMDC, DiDeMo, VATEX, YouCook2, ActivityNet Caption, TGIF, TVC, MSCOCO, Flickr30K and VQAv2. Train/val/test splits of different benchmarks of those datasets are presented in Table 11.

Specifically, **MSRVTT** (Xu et al., 2016) dataset consists of 10K video clips and 20K captions. Our evaluation encompasses text-to-video retrieval, video captioning, and video QA tasks on this dataset. For retrieval, we adopt the '1K-A split' and for captioning and QA, we adhere to the standard split. **MSVD**(Chen & Dolan, 2011) dataset comprises 1,970 videos, each accompanied by approximately 40 captions. Our evaluation involves video captioning using the official split and video QA using the split proposed by Xu et al.(Xu et al., 2017). **LSMDC** (Rohrbach et al., 2017) dataset consists of 118,000 clips from 202 movies, with each clip paired with a single caption. We evaluate text-to-video retrieval on this dataset using the official split. **DiDeMo** (Anne Hendricks et al., 2017) dataset includes 10K long-form videos sourced from Flickr. For each video, four short sentences are annotated in temporal order. We concatenate these short sentences and evaluate 'paragraph-to-video' retrieval and the official split is employed. **VATEX**(Wang et al., 2019) dataset contains 41,250 video clips sourced from the Kinetics-600 dataset(Kay et al., 2017), accompanied by 825,000 sentence-level descriptions. Our evaluation focuses on text-to-video retrieval and video captioning tasks. Regarding captioning, we employ the official split, while for retrieval, we follow the HGR (Chen et al., 2020) split protocol. **YouCook2** (Zhou et al., 2018) dataset comprises 14K video clips extracted from 2K instructional cooking videos on YouTube. Each video showcases multiple actions performed by a chef, along with corresponding textual descriptions and temporal annotations. We assess video captioning on this dataset using the official splits. **ActivityNet Caption**(Krishna et al., 2017a) dataset encompasses 20K long-form videos sourced from YouTube whose average length are 180 seconds. We evaluate text-to-video retrieval and video QA on this dataset. For retrieval, we use the official split, while for video QA, we adopt the split proposed by Yu et al.(Yu et al., 2019). **TGIF** (Jang et al., 2017) dataset features three video QA benchmarks, including TGIF-Action, TGIF-Transition, and TGIF-Frame, with the first two being multiple-choice QA and the last one being open-ended

Table 11: Downstream datasets splits.

| Task | Benchmark | #Videos/#Images | | | #Captions/#QA-pairs | | |
| --- | --- | --- | --- | --- | --- | --- | --- |
| | | Train | Val | Test | Train | Val | Test |
| Text-to-Video Retrieval | MSRVTT | 9000 | - | 1000 | 180000 | - | 1000 |
| | DiDeMo | 8394 | 1065 | 1003 | 8394 | 1065 | 1003 |
| | ANET | 10009 | - | 4917 | 10009 | - | 4917 |
| | LSMDC | 101046 | 7408 | 1000 | 101046 | 7408 | 1000 |
| Video Captioning | MSRVTT | 6513 | 497 | 2990 | 130260 | 9940 | 59800 |
| | YouCook2 | 10337 | 3492 | - | 10337 | 3492 | - |
| | MSVD | 1200 | 100 | 670 | 48774 | 4290 | 27763 |
| | VATEX | 25991 | 3000 | 6000 | 259910 | 30000 | 60000 |
| | TVC | 86603 | 10841 | - | 174350 | 43580 | - |
| Video QA | MSVD-QA | 1200 | 250 | 520 | 30933 | 6,415 | 13157 |
| | TGIF-FrameQA | 32345 | - | 7132 | 39389 | - | 13691 |
| | MSRVTT-QA | 6513 | 497 | 2990 | 158581 | 12278 | 72821 |
| | ANET-QA | 3200 | 1800 | 800 | 32000 | 18000 | 8000 |
| Text-to-Image retrieval | MSCOCO | 113287 | 5000 | 5000 | 566747 | 25010 | 25010 |
| | Flickr30K | 29000 | 1014 | 1000 | 145000 | 5070 | 5000 |
| Image Captioning | MSCOCO | 113287 | 5000 | 5000 | 566747 | 25010 | 25010 |
| Image QA | VQAv2 | 82783 | 40504 | 37K/81K | 4437570 | 2143540 | 1.1M/4.5M |

QA. We evaluate VAST on the TGIF-Frame benchmark using the official split. **TVC** (Lei et al., 2020) dataset is a multi-channel video captioning dataset, comprising 108,000 video moments and 262,000 paired captions. Additional input in the form of video subtitles can be utilized. We evaluate video captioning on this benchmark using the official split with vision information utilized only. **MSCOCO** (Lin et al., 2014) dataset contains 123K images, each paired with five annotated captions. We evaluate text-to-image retrieval and image captioning tasks on this dataset using the Karpathy split (Karpathy & Fei-Fei, 2015). **Flickr30K** (Plummer et al., 2015) dataset consists of 31K images, with each image paired with five annotated captions. We assess text-to-image retrieval on this dataset using the Karpathy split (Karpathy & Fei-Fei, 2015). **VQAv2** (Goyal et al., 2017) dataset comprises 1.1 million questions and 11.1 million answers related to MSCOCO images and we utilize train+val set for training and evaluate on the online test server.

### A.3 VISUALIZATIONS OF DIFFERNET SAMPLING METHODS

In our experiments, COSA trained with random sampling works better than the one trained with relevant sampling. As shown in figure 5, we observe a real video-language sample from ActivityNet caption dataset, which is shown in the top green part of figure, we can find that the 1) sampled vision frames are relevant and coherent, and 2) the corresponding texts are very informative and can strictly match to different frames in temporal order. Motivated by this, an ideal large-scale pretraining dataset for video-language tasks should also possess those two characteristics to help model learn the capabilities of perceiving evnets happed in videos and their relative order. In this work, we propose COSA to imitate those pretrain dataset from image-text data. And the examples of COSA with different sampling methods are shown in the bottom pink part of the figure, from which we can find that even though the random sampling example shows little relevance between four independent scenes, but it has the strenth that each sentence can strictly correspond to one image, which satisfy the second charatersic. By contrast, even though relevant (vision/text similarity) sampling examples have shown vision relevance to some extent, they have strong semantic redundancy, and one sentence can correspond to many frames, which could confuse models' event-level temporal learning. And we think that's why it performs weaker than random sampling. In addition, considering that the scene changes are somewhat hard in random sampling, we think other soft sampling methods could be a future research direction.

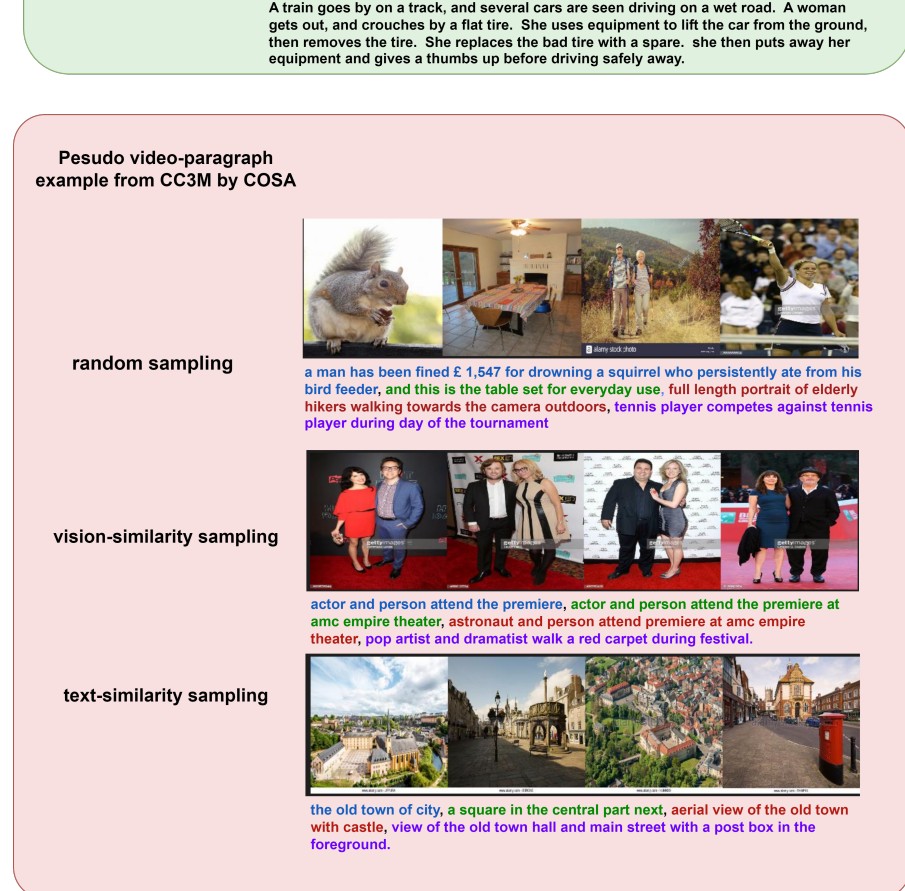

Figure 5: Visualization of different sampling methods of COSA. The top green part in the figure is the example from real video dataset (ActivityNet-captions), and bottom pink part in the figure are examples of COSA with different sampling methods.

Table 12: Downstream task finetuning settings of COSA models. Lr, Bs, Epo, F_train, F_test and Res denote learning rate, batch size, epoch, training sampling frames, testing sampling frames and resolution, respectively.

| Task | Benchmark | Lr | Bs | Epo | F_train | F_test | Res |
|---|---|---|---|---|---|---|---|
| Text-to-Video Retrieval | MSRVTT | 2e-5 | 64 | 3.6 | 8 | 16 | 224 |
| | VATEX | 2e-5 | 64 | 2.5 | 8 | 16 | 224 |
| | DiDeMo | 2e-5 | 64 | 40 | 8 | 32 | 224 |
| | ANET | 2e-5 | 64 | 20 | 8 | 32 | 224 |
| | LSMDC | 2e-5 | 64 | 5 | 8 | 32 | 224 |
| Video Captioning | MSRVTT | 2e-5 | 128 | 10 | 8 | 8 | 224 |
| | YouCook2 | 3e-5 | 64 | 30 | 8 | 16 | 224 |
| | MSVD | 1e-5 | 64 | 1.2 | 8 | 8 | 224 |
| | VATEX | 2e-5 | 64 | 10 | 8 | 20 | 224 |
| | VATEX(SCST) | 7e-6 | 64 | 5 | 8 | 20 | 224 |
| | TVC | 3e-5 | 64 | 40 | 8 | 8 | 224 |
| Video QA | MSVD-QA | 1e-5 | 64 | 8 | 8 | 18 | 224 |
| | TGIF-FrameQA | 2e-5 | 64 | 10 | 4 | 4 | 224 |
| | MSRVTT-QA | 2e-5 | 64 | 4.5 | 8 | 8 | 224 |
| | ANET-QA | 2e-5 | 64 | 10 | 8 | 16 | 224 |
| Text-to-Image retrieval | MSCOCO | 1e-5 | 256 | 5 | - | - | 384 |
| Image Captioning | MSCOCO | 1e-5 | 64 | 5 | - | - | 480 |
| | MSCOCO(SCST) | 2.5e-6 | 64 | 2.5 | - | - | 480 |
| Image QA | VQAv2 | 2e-5 | 128 | 20 | - | - | 384 |

## A.4 FINETUNING SETTINGS

Specific finetuning settings of COSA model such as learning rate, batch size, training epochs and sampled frames are presented in Table 12.

## A.5 QUALITATIVE COMPARISON ON VIST BENCHMARK

The qualitative comparisons between model trained with SST (Single Sample Training) and proposed COSA (Concatenated Sample Training) prediction on visual story telling benchmark (VIST) is shown in Figure 6, from which we can find that COSA can generate comprehensive captions (storys) for multiple frames under both zero-shot and finetuning settings, while SST can only generate one sentence describing limited contents under zero-shot setting, and is easy to generate duplicated captions under finetuning settings.

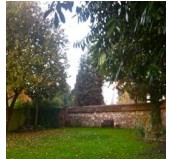 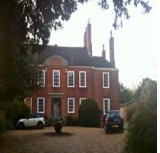 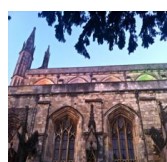 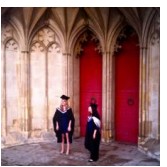 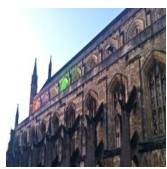

*SST(ZS)* : english civil parish in the sunshine.

*SST(FT)* : this is where i grew up. this is where i grew up. this is where i grew up. this is where i grew up. this is where i grew up. this is where i am growing up.

*COSA(ZS)* : the front of the house. a view of the cathedral from outside the cathedral. a photo of a graduate and a student in a graduation ceremony. a city in the sunshine.

*COSA(FT)* : the lawn was very nice. the building was very nice. the architecture was very old. the students were very proud of themselves. the building was very beautiful.

*GroundTruth* : this is where i go to school. i like the building a lot because it looks nice. at this angle it looks even better. a couple of my classmates are going to graduate. this is one beautiful building .

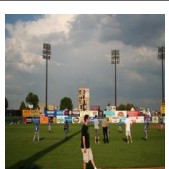 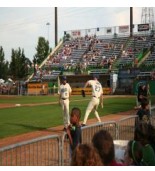 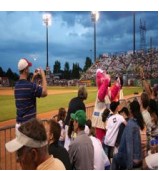 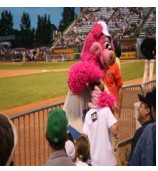 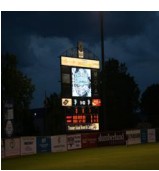

*SST(ZS)* : person, center, watches a game.

*SST(FT)* : we went to the baseball game. there were a lot of people there. some of them were crazy. some of the players didn't like it. the crowd was crazy.

*COSA(ZS)* : baseball players warm up before a game. fans watch a baseball game. person, the mascot, is shown during a game against baseball team. a general view of the scoreboard.

*COSA(FT)* : i went to the baseball game last weekend. there were a lot of people there. i took a lot of pictures. i got to meet some of the players. it was a great night.

*GroundTruth* : we are going to watch a baseball game today. the players are doing a great job. we even got to interact with the crowd. the mascot was great , and very funny. final score is here , what a great day.

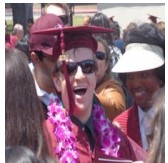 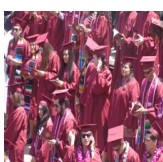 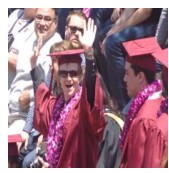 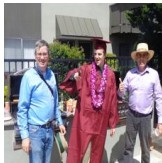 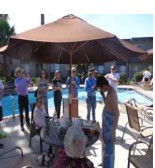

*SST(ZS)* : a graduate celebrates after receiving his diploma.

*SST(FT)* : he was excited to finally graduate from college. the group of friends and family gathered in the pool for the graduation ceremony. the boy was very excited to finally graduate. the family was very proud of him.

*COSA(ZS)* : person is all smiles after receiving his diploma during the graduation ceremony. person, waves to family and friends during the graduation ceremony. person, left, and person are all smiles after receiving his diploma at the school's commencement ceremony. the pool is full of people.

*COSA(FT)* : graduation day is finally here. all of the students line up to receive their diplomas. the students wave to the crowd. family and friends pose for a picture. after the ceremony, the family relaxes by the pool.

*GroundTruth* : graduation day has finally arrived. all the students started filing in. he was thrilled to finally be graduation. everyone posed for pictures outside the venue. and to cap off the day, we all hung out at the pool .

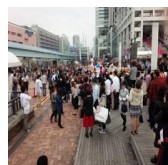 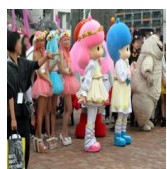 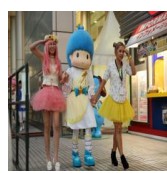 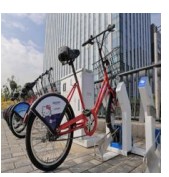 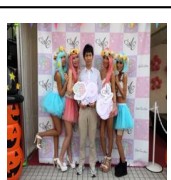

*SST(ZS)* : a group of people pose for a photo.

*SST(FT)* : there were a lot of people at the event today. the characters were very popular. the characters were very popular. the characters were very funny. the characters were very popular.

*COSA(ZS)* : people walking in the street. person, left, and person, right, enjoy a day out. a bicycle with a basket attached to it. person with the girls at the party

*COSA(FT)* : there were a lot of people at the festival today. there were a lot of funny characters. some of them were very cute. they rode their bikes around the city. it was a great time.

*GroundTruth* : the streets were filled during the festival. some came dressed in huge cartoon outfits. they were fun to dance and take pictures with. we got around using rent-able bikes. it was a memorable day.

Figure 6: Qualitative comparison between SST and COSA on the visual story telling (VIST) benchmark. ZS and FT denote zero-shot and finetuning, respectively.

