# OpenReview forum: "COSA: Concatenated Sample Pretrained Vision-Language Foundation Model"
_ICLR.cc/2024/Conference — ICLR 2024 poster_

### Official Review · Reviewer_bdF4 · 2023-10-20

**Soundness:** 3 good
**Presentation:** 3 good
**Contribution:** 3 good
**Rating:** 8
**Confidence:** 3

**Summary:**

This paper proposed a concatenated sample pretrained vision-language foundation model. By sequentially connecting multiple image-text pairs as pre-training inputs, it can jointly model visual content and event-level temporal cues using only image-text corpora. Extensive experiments show the effectiveness of the proposed method.

**Strengths:**

1.	Motivations: This paper presents a very important problem of how to capture time-level event clues using only image-text data. In the case of insufficient quality and quantity of video data, it provides a very important help for the pre-training of vision-language foundation model. The proposed method is very simple and effective. I think this work is easy to follow and most of the techniques are correct.

2.	Extensive experiments: A large amount of experimental evidence is provided in this paper, which fully verifies the effectiveness of the proposed method.

**Weaknesses:**

1.	Technical contributions: The proposed method is simple and effective. However, the proposed method is not surprising enough, because using pictures to enhance video pre-training has been quite explored in the field of CV/ vision-language pre-training field. This paper combines many existing pre-training methods, so the technology sharing is limited.

2.	More explanation: The continuous frames in the video are similar, and there are relations between different events. Establishing event-level correlation includes two parts: (1) the first part is to distinguish between different events. (2) the second part is to make temporal inferences between similar or related frames. The proposed method randomly splices several pictures, but there is no correlation between these pictures, so the model can only distinguish different events, but can not make the model time sequence inference between frames. Therefore, I do not think that the proposed method fully corresponds to its motivations.

**Questions:**

1. In Table 9, why is it better to concatenate random images for training than to concatenate only semantically similar images?
2. It is better to give the weights of the 6 losses (training objectives) and the size of the input image in the implementation details.

---

> ### Author Response · Authors · 2023-11-18
> **Response to Reviewer bdF4**
>
> ## **Q1:  Using pictures to enhance video pre-training has been quite explored in the field of CV/ vision-language pre-training field**
> ## **A1:**
>  Indeed many video-language pretraining methods also using image-text data (CC3M) as their training corpora, **but they view each picture as individual training sample.** By contrast, **COSA is the first work to utilize pictures to compose pesudo video-paragraph training data and proved that can  further improve performances of series of understanding and generation tasks**. In other words, we believe that our technical contribution lies in COSA's proposal of a novel approach to effectively utilize image data for video-language pretraining, rather than just employing both image and video data in pretraining, which has already been widely explored.
>
> ## **Q2: Proposed method does not fully correspond to its motivations.**
> ## **A2:**
>  We also agree on your analysis of two parts of temporal modeling, i.e., inter-scene temporal correlation and intra-scene temporal correlations. However, **as wrote in the "Temporal Learning in Video-Language Pretraining (VidLP)" section of Related Work** in the paper, we have attempted to let **"event-level temporal" refer explicitly to the first part, i.e. inter-scene temporal, instead of both parts**, and we use "action-level temporal" to refer to the second part, i.e. intra-scene temporal.  There are some works like OmniVL and HiTeA tried to model action-level temporal during pretraining, while COSA emphasizes more at event-level temporal learning, considering that action-level (intra-scene) needs consecutive images in a scene, which is hard to realize using pure image-text data. We also think that strengthening both action-level and event-level temporal modeling in a unified framework could be a meaningful future video-language research direction. **The motivation of current version of COSA  is to strength "event-level" (first part) temporal modeling only instead of both "action-level" (second part) and "event-level" temporal modeling.** **And we have made the illustration more clear in the updated version of Related Work**.
>
> ## **Q3: Why is it better to concatenate random images for training than to concatenate only semantically similar images**
> ## **A3:**
> In our experiments, COSA trained with random sampling works better than the one trained with relevant sampling. **We make visualizations of different sampling methods and add explanations in the updated version of supplementary materials.**
>
> Specifically, as figure 1 shown in the supplementary material, when we observe a real video-language sample from ActivityNet caption dataset (the top green part of figure), we can find that the 1) sampled vision frames are relevant and coherent, and 2) the corresponding texts are very informative and can strictly match to different frames in temporal order.
>
> Motivated by this, an ideal  large-scale  pretraining dataset for video-language tasks should also possess those two characteristics to help model learn the capabilities of perceiving events happend in videos and their relative order.  In this work, we propose COSA to imitate those pretrain dataset from image-text data. And the examples of COSA with different sampling methods are shown in the bottom pink part of the figure, from which we can find that even though the random sampling example shows little relevance between four independent scenes, but it has the strength that each sentance can strictly correspond to one image, which satisfies the second charatersic.
>
> By contrast, even though relevant (vision/text similarity) sampling examples have shown vision relevance  to some extent, they have strong semantic redundancy, and one sentence can correspond to many frames, which could confuse models' event-level temporal learning. And we think that's why it performs weaker than random sampling. In addition, considering that the scene changes are somewhat hard in random sampling, we think other soft sampling methods could be a future research direction.
>
> ## **Q4: It is better to give the weights of the 6 losses (training objectives) and the size of the input image in the implementation details.**
> ## **A4:**
> We haven't intended to tune the 6 training losses, and they  are simply equally weighted. With regards to input image size, as presented in Table 3 of Appendix, for pretraining and video tasks finetuning, we use 224 resolution, and for image-text tasks finetuning, we use 384 for QA and retrieval, and 480 for captioning. **We have added loss weights and input image size to the implementation details at the updated version of paper, thanks for your advice**.

---

> ### Comment · Reviewer_bdF4 · 2023-11-22
>
> I appreciate the responses given by the authors. The newly conducted experiments and visualizations have addressed my concerns. I would like to keep my positive rating.

---

### Official Review · Reviewer_gWLj · 2023-10-29

**Soundness:** 2 fair
**Presentation:** 3 good
**Contribution:** 2 fair
**Rating:** 6
**Confidence:** 3

**Summary:**

This paper proposed a new vision-language pre-training framework, called COSA. In particular, COSA augmented original image-text pairs by concatenating multiple examples as pseudo video-text pairs. Extensive experiments were conducted covering both video-language and image-language tasks, and demonstrated the effectiveness of the proposed method.

**Strengths:**

- The paper is well written and easy to follow. In addition, the proposed method was supported by comprehensive experiments together with ablation studies, which made the paper a complete work.
- The method COSA itself was simple yet effective to improve the learned representations for downstream tasks, and at the same time, it did not introduce extra computational costs.

**Weaknesses:**

- The method was more like a trick of data augmentation instead of a significant technical contribution, as it just simply concatenated images and their corresponding captions and it was not very surprising to observe performance improvements.
- As it was mentioned in the paper that apart from modified objectives, COSA also included original objectives for pre-training on image-text pairs. It was a complicated design to have so many training objectives and it was unclear how they were weighted (seemed to be equally weighted). Even though there was an ablation study of training objectives in Table 7, it still did not explain well the contributions of each item.
- The method leveraged the average pooled [CLS] token for each image as the final representation for the pseudo video. In this way, there was actually no temporal information considered. And the selected downstream tasks were less dependent on temporal information in the meanwhile. It would be better if tasks such as temporal action localization were included to show whether COSA can improve those tasks. In addition, since temporal information did not play any role in current method, I am afraid that using augmentations like mixup for videos/images might lead to similar performance gain, as shown in [1].
- Previous works showed that using CLIP initialization could lead to better performance. Among compared baseline methods, some of them such as MILES [2] actually used ViT trained on ImageNet for image classification and it was not a fair comparison to COSA with CLIP initialization.

[1] Hao, Xiaoshuai, et al. "Mixgen: A new multi-modal data augmentation." Proceedings of the IEEE/CVF Winter Conference on Applications of Computer Vision. 2023.

[2] Ge, Yuying, et al. "Miles: Visual bert pre-training with injected language semantics for video-text retrieval." European Conference on Computer Vision. Cham: Springer Nature Switzerland, 2022.

**Questions:**

- Is it possible for the authors to include tasks which rely much on temporal information like temporal action localization? This would provide better understanding of the proposed method.
- It would be better if results with different initializations can be presented to remove my concern about better CLIP initialization.
- It was worth trying data augmentations like mixup and it might lead to similar performance gain as demonstrated in the paper.

---

> ### Author Response · Authors · 2023-11-18
> **Response to Reviewer gWLj - Part1**
>
> ## **Q1: Novelty and comparison with other data augmentation methods.**
> ## **A1:**
> - We think viewing COSA from the data augmentation perspective also makes sense, considering that the core operation of COSA is to concatenate image-text data into pseudo video-paragraph data, and learn event-level temporal correspondences which are needed for video-text downstream tasks. **However, as far as we know, we are the first to compose pseudo video-paragraph data from image-data, demonstrating its effectiveness across abundant both image-text and video-text tasks, making detailed comparison with the common SST baseline, and verifying its generality across different dataset and model scales, so we think COSA has its novelty.**
>
> - In addition, as suggested, we add experiments to make comparisons with other data augmentation method. Mixup is first proposed for image classification tasks which sample two images and interpolate both raw images and their labels, in equal weight. Motivated by Mixup, Mixgen promotes it from image field to image-text field, replacing labels with captions and keeping image processing remained (average raw pixels of two images). We add the comparison results as follows, in which we find that compared to SST baseline, mixgen severely decreases the performance of downstream video-text tasks, while COSA improves SST baseline evidently. We attribute this phenomenon to that Mixgen simply averages different images into one single image without any temporal modaling, leaving event-level temporal learning unconsidered, thus the more images averaged (4 vs 2), the more performance degrades. By contrast, COSA concatenates multiple images  into videos and encodes temporal information via temporal position embeddings, which can effectively learn event-level temporal correspondence, and more images can further improve performances. These results demonstrate that COSA is more general and effective compared to Mixgen.
> | Method | # Concatenated or mixed Images | DiDeMo-RET (R@1) | ActivityNet-RET (R@1) | VIST-CAP(CIDEr) | TVC-CAP(CIDEr) | TGIF-QA(Acc) |
> | --- | --- | --- | --- | --- | --- | --- |
> | SST | - | 49.2 | 43.8 | 14.8 | 53.9 | 73.0 |
> | COSA(Ours) | 2 | 49.1 | 45.5 | 15.3 | 54.5 | 73.5 |
> | COSA(Ours) | 4 | 54.8 | 51.2 | 20.7 | 55.4 | 73.7 |
> | Mixgen | 2 | 46.5 | 42.9 | 11.9 | 53.6 | 73.1 |
> | Mixgen | 4 | 40.8 | 37.2 | 10.0 | 52.9 | 72.3 |
>
> ## **Q2: Different model initialization beyond CLIP.**
> ## **A2:**
> - In fact, for fair comparison with state-of-the-art methods, we train different scales of COSA model with different vision encoders and training corpus. As depicted in the following table (the same as Table 1 in the paper), COSA-B takes **Swin-Base** trained on ImageNet as vision encoder, **instead of CLIP** pretrained on 400M WebDataset.
> | Model |  Vision Encoder |
> | --- | --- |
> | COSA-B | Swin-B |
> | COSA-L | CLIP/ViT-L/14 |
> | COSA |  EVAClip/VIT-G/14 |
>
> - **We copy the comparison results of COSA-B from Table 2 in the paper below, and additionally add a vision encoder column**. It is noted that **all methods use vision backbones pretrained on ImageNet, and thus we believe a fair comparison is warranted.** Methods using Swin as backbone generally outperform the ones using ViT as backbone, and COSA surpasses other methods like Clover, VIOLETv2, LF-VILA, which also use Swin. Considering that Swin may be marginlly better than ViT backbone, we are training a COSA-B(ViT) model for a absolute fair comparison with methods initialized with ViT, such as MILES. However, this experiment requires a longer training period compared to our other experiments conducted for this rebuttal, due to the increased number of training iterations and the non-fixed vision backbone. Consequently, we are unable to complete it before the end of the rebuttal period. We will add this comparison results at the final version of paper.
> | Method | Vision encoder | Example | MSRVTT | DiDeMo | LSMDC | ActivityNet |
> | --- | --- | --- | --- | --- | --- | --- |
> | ClipBert | ResNet50 | 5.4M | 22.0 /46.8/59.9  | 20.4/48.0 /60.8 | - | 21.3/49.0/63.5 |
> | Frozen | ViT-B | 5M | 31.0/59.5/70.5  | 34.6/65.0/74.7 | 15.0/30.8 /39.8 | - |
> | BridgeFormer | ViT-B | 5M | 37.6/64.8/75.1 | 37.0/62.2/73.9 | 17.9/35.4/44.5 | - |
> | MILES | ViT-B | 5M | 37.7/63.6/73.8 | 36.6/63.9/74.0  | 17.8/35.6/44.1  | - |
> | OA-Trans | ViT-B | 5M | 35.8/63.4/76.5 | 34.8/64.4/75.1 | 18.2/34.3/43.7 | - |
> | Clover | Swin-B | 5M | 38.6/67.4/76.4 | 45.1/74.3/82.2 | 22.7/42.0/52.6 | - |
> | VIOLETv2 | Swin-B  | 8.5M | 37.2/64.8/75.8  | 47.9/76.5/84.1  | 24.0/43.5/54.1 | - |
> | LF-VILA | Swin-B | 5M |-  | 35.0/64.5/75.8 | - | 35.3/65.4/- |
> | COSA-B | Swin-B | 5M | **42.2/69.0/79.0** | **57.8/80.6/87.9** | **27.3/46.7/55.2** | **55.6/80.7/89.1**  |

---

> > ### Comment · Reviewer_gWLj · 2023-11-22
> >
> > Thanks for your detailed response. I acknowledge that the paper demonstrated a simple and effective method to improve video-language pre-training empirically but the technical contribution is limited. Considering the completeness of the paper, I would increase my score to 6.

---

> ### Author Response · Authors · 2023-11-18
> **Response to Reviewer gWLj - Part2**
>
> ## **Q3: Evaluation on temporal action localization benchmark.**
> ## **A3**:
>  - We evaluate COSA and SST on **temporal action localization (TAL)** task. Specifically, we take **actionformer** as baseline methods and evaluate on **ActivityNet dataset** with features extracted from vision encoders of COSA-B and SST-B respectively. It is noted that they are both trained with the same datasets and iterations. The results are shown in the following table, from which we can find that both SST and COSA improve performance over baseline. Furthermore, COSA surpasses SST, demonstrating its efficacy on the localization task.
> | Method | Vision feature | mAP |
> | --- | --- | --- |
> | Actionformer | Swin-B | 33.40 |
> | Actionformer | SST-B | 34.76 |
> | Actionformer | COSA-B | 35.10 |
>
> - However, the improvements appear limited, which we believe may be due to the fact that COSA is pre-trained as an integrated vision-language framework, but when fine-tuned on TAL, only its vision encoder is utilized. This approach might not fully leverage the strengths of the COSA framework. So we additionally evaluate COSA on **text-guided video grounding task** (i.e., moment retrieval).
> Specifically, inspired by **MMN**, we sparsely sample 16 frames per video and forward them to the vision encoder,  getting a 1d feature map whose size equals to 16\*C (C is the hidden size of channel). After that we transform the 1d feature map to 2d map whose size is 16\*16*C. Each feature located at (i, j) in the feature map represents a pre-defined anchor with fixed start and end timestamps (which are the timestamps of i-th and j-th frames), and they are computed by averaging the features of i-th and j-th frames along channel. It is noted that only the upper triangle part of 2d map is valid due to that start time must be smaller than end time. Input querys are processed by the text encoder of COSA.  The whole training losses consist of contrastive loss and IOU loss, following MMN (constrastive head is inherited from pretrained COSA model and IOU head is newly initialized). All parameters of networks are updated during finetuing. We conducted experiments on the **Charades dataset**. As presented in the following table, COSA-B surpasses SST-B with large margins on the R@1 metric of different IOU scores, strongly proving that COSA can also benefit localization tasks.
> | Method | R@1_IOU0.3 | R@1_IOU0.5 | R@1_IOU0.7 |
> | --- | --- | --- | --- |
> | Baseline(w/o pretrain) | 67.69 | 54.49 | 32.15 |
> | SST-B | 68.66 | 56.45 | 34.84 |
> | COSA-B | 69.62 | 58.01 | 36.40 |
>
> ## **Q4: More explanations of training losses.**
> ## **A4:**
> Since there are up to six losses (ITC/ITM/CITC/CITM/CMLM/CGM) used in COSA, we do not tune their weights considering the complexity. Using identical weights already achieves good performance. We also make experiments to analyze the function of each loss. For clarity, we group ITC and ITM into L_align (L_align=L_itc+L_itm), and group MLM and GM into L_mask (L_mask = L_mlm+L_gm). The analysis is as follows:
> - ITC improves retrieval task (line1 vs line0), and ITM make further improvements via fine-grained multimodal fusion and rereanking (line2 vs line1).
>  - MLM (bi-direction self-attention masks) can improve caption and QA tasks (line3 vs line0), and adding GM (causal self-attention mask) can further enhance performance (line4 vs line3).
> -  Using L_align and L_mask together leads to notable improvements on all tasks (line5 vs line4).
> -  Regarding the 'Concatenate' operation in COSA, replacing L_mask with L_Cmask and incorporating L_Calign both contribute to improved performance across all tasks (line6 vs line5, line7 vs line6).
>
> | Line | Losses  | DiDeMo-RET | MSRVTT-RET | TVC-CAP | TGIF_QA |
> | --- | --- | --- | --- | --- | --- |
> | 0 | - | 26.9 | 34.1 | 51.0 | 69.4 |
> | 1 | L_itc | 38.4 | 37.1 | 49.7 | 66.6 |
> | 2 | L_align (L_itc+L_itm) | 48.4 | 38.3 | 50.3 | 69.2 |
> | 3 | L_mlm | 26.0 | 31.6 | 52.1 | 72.1 |
> | 4 | L_mask (L_mlm+L_gm)  | 30.9 | 33.5 | 53.3 | 72.9 |
> | 5 | L_align+L_mask | 49.2 | 39.4 | 53.9 | 73.0 |
> | 6 | L_align+L_Cmask | 53.0 | 42.8  | 55.7  | 73.7 |
> | 7 | L_align+L_Calign+L_Cmask | 54.8  | 43.5  | 55.4 | 73.7 |

---

> ### Author Response · Authors · 2023-11-18
> **Response to Reviewer gWLj - Part3**
>
> ## Reference
> - Hao et al. Mixgen: A new multi-modal data augmentation.
> - Zhang et al. Actionformer: Localizing moments of actions with transformers
> - Wang et al. Negative Sample Matters: A Renaissance of Metric Learning for Temporal Grounding

---

### Official Review · Reviewer_vqzh · 2023-10-30

**Soundness:** 2 fair
**Presentation:** 2 fair
**Contribution:** 2 fair
**Rating:** 5
**Confidence:** 5

**Summary:**

The work proposes the vision-language foundation model, which can jointly model visual contents and event-level temporal cues using only image-text corpora. Extensive experiments demonstrate that COSA consistently improves performance across a broad range of semantic vision-language downstream tasks.

**Strengths:**

1. The paper proposes the effective method for video-text and image-text tasks.
2. The experiment is very adequate.  The model consistently improves performance
across a broad range of semantic vision-language downstream tasks.

**Weaknesses:**

1. The reasons for the improvement brought by Concatenation lack detailed analysis. Why is there also improvement for image-text tasks? Why is it necessary to include the video dataset (web2vid)? Why wasn't the 1.2B model included in the video dataset?
2. The data shown in Table 1 is confusing. The data for COSA-L is 417M, while the data volume for COSA is 415M.
3. The results in Table 7 and Table 7 are also confusing. The best performance is based on 6 pretraining task? Which pre-training tasks were used in the overall experimental results of COSA? Is the WebVid2.5M dataset more important，the results for COSA 4 frames?

**Questions:**

see weaknesses

---

> ### Author Response · Authors · 2023-11-18
> **Response to Reviewer vqzh - Part1**
>
> ## **Q1: Why is there also improvement for image-text tasks?**
> ## **A1:**
> Through randomly concatenating image-text or video-text samples, COSA can learn explicit and accurate event-level temporal relations through large-scale pretraining, and this brings improvements on downstream video-language understanding tasks. With regard to the improvements for image-text benchmarks as shown in Table 6 of main paper (COCO retrieval, COCO caption and VIST story telling), we explain from the following perspectives.
> - **Learning difficulty**. Building the relations of image objects and text tokens are the core of image-text pretraining, the stronger relations have been built, the better models can handles cross-modality tasks. During pretraining, SST (single sample training) gives model one image and one sentence once, and model needs to connect objects and textual words. By contrast, COSA give models multiple images and sentences, and the numbers of both objects and words become around four times more (if four samples are concatenated). This makes the relations learning process harder, because for one object, the positive concepts remains the same while negative words becomes more. Similarly, for a meaningful word, there are four times more negative visual objects. The relation building process becomes more difficult, which may benefit in that learned features and relations are more robust and discriminative.
> - **Data augmentation**. During SST training, each image-text pair is seen multiple times (=epochs). By contrast, For COSA, when the concatenation number is equal to 4, with a larger batch size and random sampling, there is negligible possibility that concatenated samples are the same, which means at every iteration, the input cross-modality pairs are different, which could be viewed as a kind of data augmentation to avoid model overfitting.
>
> ## **Q2:  Necessity of including the video dataset (web2vid).**
> ## **A2:**
> - In abalation study (Table 8), we include the video dataset for two purposes. 1) Proving the generality of COSA method, which could be fitted to both image-text and video-text data. 2) Proving applying COSA method to video dataset works better than SST baseline with sampling more frames, further demonstrate the strength of COSA.
> - In SOTA comparisons (COSA-B and COSA-L), they are trained with video dataset, mainly for **fair comparison**, because most compared methods also train their model on both image and video datasets.
> - In SOTA comparisons (COSA), it is trained without using any video dataset. It is noted that we assume adding video datasets like webvid could further improve COSA's performance, but we want to convey that **even using image-text data only**, we can still outperform other large models using video datasets (videoCoCa, Flamingo) or other models also using image data only (GIT), and that is **aligned with COSA's motivation to relief the scarcity of high-quality video data**, by using concatenated image-text samples.

---

> ### Author Response · Authors · 2023-11-23
> **Response to Reviewer vqzh - Part2**
>
> ## **Q3: Training data volume of COSA-L and COSA is confusing.**
> ## **A3:**
> Sorry for the confusion caused. For clarity, the details of the data used in training COSA/COSA-L are listed in the table below. The difference in data volume is due to the **exclusion of the Webvid-2.5M dataset in the COSA training**. The reason of this exclusion is explained in our response (A2) to "Q2: Necessity of including the video dataset (Web2Vid)". As noted in the caption of Table 1, since the LAION-102M used in COSA is sampled from the training corpus of the EVAClip vision backbone (LAION-400M), we count them collectively as 400M.
>
> | Model | Vision-text pairs utilized in vision-language pretraining | Vision-text pairs utilized in vision encoder pretraining | Total |
> | --- | --- | --- | --- |
> | COSA-L | CC14M (**15M**) + Webvid2.5M(**2.2M**)  | Webdata400M(**400M**) | **417M** |
> | COSA | CC14M (**15M**) + LAION102M(102M, overlapped with LAION400M) | LAION400M(**400M**) | **415M** |
>
> - "CC14M" combines the CC3M, CC12M, COCO, VG, and SBU caption datasets, and after the removal of invalid links, totaling 15M  pairs remain accessible.
>
> - "Webvid2.5M" initially contains 2.5M vision-text pairs, and after filtering invalid links, we achieve 2.2M pairs for training.
>
> ## **Q4:  The results in Table 7 and Table 8 are also confusing. The best performance is based on 6 pretraining task? Which pre-training tasks were used in the overall experimental results of COSA? Is the WebVid2.5M dataset more important, the results for COSA 4 frames?**
> ## **A4:**
> - Yes.  **As written in Table 7 and "Training Objectives" in Section 4.3**, the best performance is based on 6 pretraining tasks, and they  are also used in all other COSA experiment tables.
> - **Table 8** targets at demonstrating COSA's generality to different datasets and backbones. We want to **emphasize the comparison between methods (i.e. SST baseline and COSA), instead of datasets (i.e., cc3m and webvid)**. **In addition, we compare webvid and cc3m in the table below, and conduct an additional experiments that use both dataset together**. From the results we can find that separately using one dataset can achieve similar performance, and using them together further improves performance on all tasks.
>
> | Method | Enc  | Dataset | Frame | DiD(R) | ANET(R) | MSR(R) |  YOU(C) | MSR(Q) |
> | --- | --- | --- | --- | --- | --- | --- | --- | --- |
> | COSA  | CLIP-B | WebVid2.5M | 1 | 54.7 | 52.2 |  44.0 | 93.4 | 46.4 |
> | COSA  | CLIP-B | CC3M | 1 | 54.8 | 51.2 | 43.5 | 91.9 | 46.5 |
> | COSA  | CLIP-B | WebVid2.5M+CC3M | 1 | 55.8 | 54.1 | 46.2 | 100.5 | 47.0 |

---

### Official Review · Reviewer_kord · 2023-11-01

**Soundness:** 3 good
**Presentation:** 3 good
**Contribution:** 3 good
**Rating:** 6
**Confidence:** 4

**Summary:**

This paper proposed to concatenate image-text samples to mimic video-paragraph corpus in vision-language pre-training. The method is simple and the evaluation is conducted on various image/video datasets to demonstrate impressive performance.

**Strengths:**

1. The idea is simple and easy to reproduce. Meanwhile, the performance gain is impressive.
2. The experiments are conducted on many benchmarks across image-text and video-text tasks, as well as different data scales. Also the ablation is comprehensive and covers most of the aspects of this method.

**Weaknesses:**

1. It makes sense that pseudo video-paragraph data in pre-training can mitigate the gap between pre-training and fine-tuning in image-text pertaining. However, intuitively, the discontinuity of semantics in pseudo video-paragraph data should hurt compared with relevant video-paragraph data because in downstream videos, image and text are indeed relevant. But in Tab9, it seems random sampling is better than relevant sampling, which is kind of counter-intuitive. Can the authors explain more about it?

2. When having seen the same number of samples, whether COSA is better than SST in `image-text downstream tasks`? Basically, I want to see the comparison like Figure 4 in image-text downstream tasks. I am okay with this observation not holding anymore in image-text downstream tasks because essentially video-paragraph and image-text are different domains.

3. I want to see how this method performs in zero-shot image-text tasks. Considering the domain gap, I suspect it might perform worse than some image-text pre-trained methods that COSA can outperform when finetuning.

**Questions:**

See weaknesses.

---

> ### Author Response · Authors · 2023-11-18
> **Response to Reviewer kord - Part1**
>
> ## **Q1: Why random sampling is better than relevant sampling?**
>
> ## **A1**:
> In our experiments, COSA trained with random sampling works better than the one trained with relevant sampling. **We make visualizations of different sampling methods and add explanations in the Figure 1 of updated version of supplementary materials**.
>
> Specifically, as Figure 1 shown in the supplementary material, when we observe a real video-language sample from ActivityNet caption dataset (the top green part of figure), we can find that the 1) sampled vision frames are relevant and coherent, and 2) the corresponding texts are very informative and can strictly match to different frames in temporal order.
>
> Motivated by this, an ideal  large-scale  pretraining dataset for video-language tasks should also possess those two characteristics to help model learn the capabilities of perceiving events happened in videos and their relative order.  In this work, we propose COSA to imitate those pretrain dataset from image-text data. And the examples of COSA with different sampling methods are shown in the bottom pink part of the figure, from which we can find that even though the random sampling example shows little relevance between four independent scenes, but it has the strength that each sentence can strictly correspond to one image, which satisfies the second characteristic.
>
> By contrast, even though relevant (vision/text similarity) sampling examples have shown vision relevance  to some extent, they have strong semantic redundancy, and one sentence can correspond to many frames, which could confuse models' event-level temporal learning. We believe that's why it performs worse than random sampling. In addition, considering that the scene changes are somewhat abrupt in random sampling, we think smoother sampling methods could be a future research direction.
>
> ## **Q2: Comparison of COSA and SST on image-text tasks when seen the same number of samples.**
> ## **A2:**
> In fact, **Figure 4 already contains two image tasks**, including MSCOCO-RET (text-to-image retrieval) and VIST-CAP (image story telling). We here represent the comparison on those two benchmarks  in table below.
>
>
> | Method    | Iteration | MSCOCO-RET (Recall@1) | VIST-CAP (Cider) |
> |-----------|-----------|-----------------------|------------------|
> | SST-30K   | 30K       | 53.9                  | 14.8             |
> | **COSA-30K**  | 30K       | 54.7                  | 20.7             |
> | SST-60K   | 60K       | 54.8                  | 18.6             |
> | COSA-60K  | 60K       | 55.3                  | 24.1             |
> | **SST-120K**  | 120K      | 54.6                  | 19.1             |
> | COSA-120K | 120K      | 55.8                  | 26.0             |
>
> From the table we can find that when comparing COSA and SST under the same examples seen setting (COSA-30K vs SST-120K), COSA-30K achieves 54.7 R@1 and 20.7 CIDEr, which outperforms SST-120K who achieves 54.6 R@1 and 19.1 CIDEr. This demonstrates that COSA is both more effective and efficient than SST (COSA-30K only takes less than half the training time of SST-120K), for both image-text and video-text tasks, showing its generality as a vision-language foundation model. It is noted that when compared under the same iteration setting (both methods are converged), COSA-120K outperforms SST-120K by a significant margin.

---

> ### Author Response · Authors · 2023-11-18
> **Response to Reviewer kord - Part2**
>
> ## **Q3: Compare COSA with state-of-the-arts on zero-shot image-text tasks.**
> ## **A3:**
> Considering MSCOCO is included in CC4M dataset which has been used  in the pretraining process of COSA, we test COSA on two image-text zero-shot benchmarks including **Flickr-30K image-text retrieval** and **Nocaps image caption**, and compare COSA with other pretraining methods.
> For Flickr-30K retrieval, we test in two settings:
> - direct zero-shot evaluation. From the results we can find that COSA outperforms methods including CLIP/ALIGN/FILIP/Florence/CoCa with evident strengths. Considering that compared methods are all two-tower models while COSA use ITM to rerank, we test COSA-ITC that removes the ITM refining process, we can find that COSA-ITC also achieves better text-to-image and decent image-to-text zero-shot performances.
> | Method   | Finetune on MSCOCO | Text-to-Image R@1/R@5/R@10 | Image-to-Text R@1/R@5/R@10 |
> |----------|--------------------|----------------------------|----------------------------|
> | CLIP     | N                  |   68.7 90.6 95.2           | 88.0 98.7 99.4             |
> | ALIGN    | N                  | 75.7/93.8/96.8             | 88.6/98.7/99.7             |
> | FILIP    | N                  |  75.0 93.4 96.3            | 89.8 99.2 99.8             |
> | Florence | N                  | 76.7/93.6/-                | 90.9/99.1/-                |
> | CoCa     | N                  | 80.4/95.7/97.7             | 92.5/99.5/99.9             |
> | COSA-ITC | N                  | 84.4/**97.2**/**98.5**            | 88.0/99.7/99.9             |
> | COSA     | N                  | **87.2**/97.0/97.9             | **96.8**/**100.0**/**100.0**           |
>
> - finetuning on MSCOCO retrieval dataset and then tested on Flickr30K. In this setting, all compared methods use ITM reranking and have an intermediate finetuning process on MSCOCO dataset. COSA surpasses ALBEF/BLIP/mPLUG-2/BLIP-2 with evident margins.
> | Method | Finetune on MSCOCO | Text-to-Image R@1/R@5/R@10 | Image-to-Text R@1/R@5/R@10 |
> | --- | --- | --- | --- |
> | ALBEF | Y | 85.6/97.5/98.9 | 95.9 99.8 100.0 |
> | BLIP  | Y | 86.7/97.3/98.7  | 96.7/100.0/100.0 |
> | mPLUG-2  | Y | 88.1/97.6/99.1 | 97.2 100.0 100.0 |
> | BLIP-2 | Y | 89.7/98.1/98.9 | 97.6/100.0/100.0 |
> | COSA (Ours) | Y | **90.2**/**98.4**/**99.4**  | **98.3**/**100.0**/**100.0**  |
>
> For the Nocaps zero-shot caption dataset, it is noted that there are models with LLM as a multimodal decoder achieving higher performance, such as BLIP-2. Considering that COSA utilizes BERT-base as its multimodal decoder, we only compare it with similar models. Specifically, we find that COSA outperforms OSCAR and VinVL by large margins and achieves comparable performance with SimVLM and BLIP.
> |  | Use LLM | Cider | Spice |
> | --- | --- | --- | --- |
> | BLIP-2 | Y | 119.7  | 15.4 |
> | OSCAR | N | 80.9  | 11.3  |
> | VinVL | N | 95.5  | 13.5 |
> | BLIP | N |  **113.2**  | 14.8 |
> | SimVLM | N |  112.2 | - |
> | COSA(Ours) | N | 113.0 | **15.0** |

---

> > ### Comment · Reviewer_kord · 2023-11-23
> >
> > Thank the author for the response. My first concern is solved.
> >
> > For the second concern, it's weird that in MSCOCO-RET, SST-120K's performance is worse than SST-60K. In the table, SST-60k already outperforms COSA-30K in MSCOCO-RET, why would SST-120K turn out to underperform COSA-30K? Explanation from the author(s) is needed here.
> >
> >  As for my third concern, I think it's better to be validated by a fair comparison such as SST-120k vs COSA-30K with the same number of seen data conducting zero-shot tasks. This is to remove the difference brought by training datasets/recipes in different works. I still think that there should be a domain shift between pre-training on random concatenated image-text and the single image-text scenario in downstream tasks if directly applying zero-shot inference. I am okay to see the negative results. This is not a weakness of this paper either.

---

> > > ### Author Response · Authors · 2023-11-23
> > > **New response to Reviewer kord**
> > >
> > > **Dear reviewer, thanks for your response.**
> > >
> > > **For the second concern**, we first analyse the reason that COSA can also improve image-text tasks from two perspectives as follows.
> > > - **Learning difficulty**. Building the relations of image objects and text tokens are the core of image-text pretraining, the stronger relations have been built, the better models can handles cross-modality tasks. During pretraining, SST (single sample training) gives model one image and one sentence once, and model needs to connect objects and textual words. By contrast, COSA give models multiple images and sentences, and the numbers of both objects and words become around four times more (if four samples are concatenated). This makes the relations learning process harder, because for one object, the positive concepts remains the same while negative words becomes more. Similarly, for a meaningful word, there are four times more negative visual objects. The relation building process becomes more difficult, which may benefit in that learned features and relations are more robust and discriminative.
> > > - **Data augmentation**. During SST training, each image-text pair is seen multiple times (=epochs). By contrast, for COSA, when the concatenation number is equal to 4, with a larger batch size and random sampling, there is negligible possibility that concatenated samples are the same, which means at every iteration, the input cross-modality pairs are different, which could be viewed as a kind of data augmentation to avoid model overfitting.
> > >
> > > Then we give the following analysis of the experiment results in the table above.
> > > - Due to that SST has lower learning difficulty and less data augmentation compared to COSA, it may converge faster than COSA and meeting overfitting earlier, and we think that's the reason **why SST-120K's performance on MSCOCO-RET is lower than SST-60K**, and COSA-120K's performance is higher than COSA-60K.
> > > - Considering that even though COSA-30K outperforms SST-120K  and COSA-120K outperforms SST-60K on the MSCOCO-RET,  COSA-30K underperforms SST-60K  on this benchmark,  as you pointed. This to some extent proves that COSA's improvement over SST on image-text tasks may be not so evident as on video-text tasks, considering that essentially video-paragraph and image-text have domain differences.
> > >
> > > **For the third concern**, we are sorry about misunderstanding your comments about zero-shot comparison between COSA and SST by comparison between COSA and other SOTA methods. As suggested, we conduct new experiments to compare COSA-30K and SST-120K on **direct zero-shot testing on Flickr-30K image-text retrieval benchmark (both ITC and ITM settings)**. As results shown in the following table, **COSA-30K outperforms SST-120K on four metrics of six under the ITC setting, and five metrics out of six  under the ITM setting**. This experiment demonstrates that under the same samples seen scenario, compared to SST, COSA has better capability to generalize to unseen dataset.
> > >
> > > | Method (ITC)  | Finetune on MSCOCO | Text-to-Image R@1/R@5/R@10 | Image-to-Text R@1/R@5/R@10 |
> > > | --- | --- | --- | --- |
> > > | SST-120K | N | 59.6/84.2/90.3 | 69.9/93.8/97.4 |
> > > | COSA-30K | N | **61.0/85.6/91.3**  | **71.6**/93.1/97.1  |
> > >
> > > | Method (ITM)  | Finetune on MSCOCO | Text-to-Image R@1/R@5/R@10 | Image-to-Text R@1/R@5/R@10 |
> > > | --- | --- | --- | --- |
> > > | SST-120K | N | 67.9/86.1/88.9 | **83.6**/96.8/98.9 |
> > > | COSA-30K | N | **69.2/87.4/89.8** | 83.5/**97.3**/**99.1** |

---

> ### Author Response · Authors · 2023-11-18
> **Response to Reviewer kord - Part3**
>
> ## Reference
> -  Lei et al. Less is more: Clipbert for video-and-language learning via sparse sampling
> - Radford et al. Learning transferable visual models from natural language supervision.
> - Jia et al.  Scaling up visual and vision-language representation learning with noisy text supervision.
> - Yao et al. FILIP: fine-grained interactive language-image pre-training.
> - Yuan et al. Florence: A new foundation model for computer vision.
> - Yu et al. Coca: Contrastive captioners are image-text foundation models.
> - Li et al. Align before fuse: Vision and language representation learning with momentum distillation.
> - Li et al.  BLIP: bootstrapping language-image pre-training for unified vision language understanding and generation.
> - Xu et al. mplug-2: A modularized multi-modal foundation model across text, image and video.
> - Li et al. Blip-2: Bootstrapping language-image pre-training with frozen image encoders and large language models
> - Li et al. Oscar: Object-semantics aligned pre-training for vision-language tasks
> - Zhang et al. Vinvl: Making visual representations matter in vision-language models
> - Wang et al. Simvlm: Simple visual language model pretraining with weak supervision.

---

### Author Response · Authors · 2023-11-18
**Rebuttal overview**

We appreciate the reviewers' positive feedback on our method, finding its **simple implementation and  reproducibility** (Reviewer kord, Reviewer gWLj,Reviewer bdF4), **impressive effectiveness** (Reviewer kord, Reviewer vqzh, Reviewer gWLj,Reviewer bdF4), **adequate ablation studies and experiments** (Reviewer kord, Reviewer vqzh, Reviewer gWLj, Reviewer bdF4) and **good writing** (Reviewer gWLj). We have carefully considered the comments and addressed all concerns in detail.

---

### Meta-Review · Area_Chair_vdAo · 2023-12-03

**Metareview:**

(a) The topic of this article is how to incorporate temporal information from video data into visual language models. The authors propose a pre-training method that leverages image-text pairs as input, but also captures the event-level temporal cues between them. The method consists of sequentially concatenating multiple image-text pairs and applying a transformer-based model to jointly learn the visual and textual representations. The article claims that this approach can enhance the performance of visual language models on various downstream tasks, such as video question answering and video captioning.

(b) The article has several strengths that make it a valuable contribution to the field of visual language modeling. First, it introduces a novel and simple pre-training method that effectively incorporates temporal information from video data into image-text pairs. Second, it conducts extensive experiments on various downstream tasks to demonstrate the effectiveness and generalizability of the proposed method. Third, it provides a comprehensive ablation study that covers most aspects of the method. The ablation study helps to explain the underlying mechanisms and design choices of the method.

(c) The shortcoming of this article is that some experimental details are still unclear, such as insufficient comparison and insufficient explanation of performance improvements. Although this method is a simple and effective one, it is slightly lacking in technical contribution.

**Justification For Why Not Higher Score:**

I did not give a higher score because its technical contribution is limited, and some key explanations are not very clear, so the conclusions do not have sufficient theoretical support.

**Justification For Why Not Lower Score:**

I didn't give it a lower score because this article does propose a simple but effective approach. The experiments are conducted on many benchmarks. Also the ablation is comprehensive and covers most of the aspects of this method.

---

### Decision · Program_Chairs · 2024-01-16

Accept (poster)